# The CLIC/GEEC pathway regulates particle uptake and formation of the virus-containing compartment (VCC) in HIV-1-infected macrophages

**Kathleen Candor[1,2], Lingmei Ding[2], Sai Balchand[2], Jason E. Hammonds[2], Paul Spearman[2]***

**1** Immunology Graduate Program, University of Cincinnati, and Infectious Diseases Division, Cincinnati Children's Hospital Medical Center, Cincinnati, Ohio, United States of America, **2** Infectious Diseases Division, Cincinnati Children's Hospital Medical Center and University of Cincinnati, Cincinnati, Ohio United States of America

* paul.spearman@cchmc.org

## Abstract

HIV-1 particles are captured by the immunoglobulin superfamily member Siglec-1 on the surface of macrophages and dendritic cells, leading to particle internalization and facilitating trans-infection of CD4+ T cells. HIV-1-infected macrophages develop a unique intracellular compartment termed the virus-containing compartment (VCC) that exhibits characteristic markers of the late endosome and is enriched in components of the plasma membrane (PM). The VCC has been proposed as the major site of particle assembly in macrophages. Depleting Siglec-1 from macrophages significantly reduces VCC formation, implying a link between the capture and uptake of external HIV-1 particles and the development of VCCs within HIV-infected cells. We found that internalization of particles to the VCC was independent of clathrin, but required dynamin-2. CD98 and CD44, classical markers of the CLIC/GEEC pathway, colocalized with Siglec-1 and HIV-1 particles within the VCC. Virus-like particles (VLPs) were taken up within CD98 and Siglec-1-enriched tubular membranes that migrated centripetally over time to form VCC-like structures. Inhibition of CLIC/GEEC-mediated endocytosis resulted in the arrest of captured HIV-1 particles on the macrophage cell surface, prevented VCC formation, and significantly reduced the efficiency of trans-infection of T cells. These findings indicate that following capture of virus by Siglec-1, particles follow an endocytic route to the VCC that requires both the CLIC/GEEC pathway and dynamin-2. We propose a model in which internalization of HIV-1 particles together with CLIC/GEEC membranes leads to the formation of the VCC in HIV-infected macrophages, creating an intracellular platform that facilitates further particle assembly and budding.

## Author summary

The major cell types infected by HIV are CD4+ T cells and macrophages. Infection of macrophages is of great interest because this cell type can contribute to transmission

**Data availability statement:** RNAseq data are available at https://www.ncbi.nlm.nih.gov/geo/query/acc.cgi?acc=GSE248999. All other relevant data are within the manuscript and its Supporting Information files, including Supporting Data Files with numeric values used to generate graphs and statistics.

**Funding:** This study was funded by R01 DA051895 to PS, from NIH, National Institute on Drug Abuse, https://nida.nih.gov/. The funders played no role in the study design, data collection and analysis, decision to publish, or preparation of the manuscript.

**Competing interests:** The authors have declared that no competing interests exist.

of virus within tissues, and because infected macrophages contribute to HIV-related complications that include neurologic disorders, endocrine disorders, and cardiovascular disease. HIV infection of macrophages may also create a latent reservoir that persists in infected individuals despite administration of antiretroviral therapy. Here we focused on a compartment that forms in HIV-infected macrophages termed the virus-containing compartment or VCC. The VCC is an intracellular compartment that has been described as a site of assembly and as a holding compartment for viruses, and contains components of both intracellular organelles and the plasma membrane. We identified an endocytosis pathway that helps to explain the origins of the VCC, the CLIC/GEEC pathway. Inhibition of the CLIC/GEEC pathway prevented virus-like particle delivery to the VCC and inhibited transmission of infection from macrophages to T cells. The GTPase dynamin-2 was also required for delivery of HIV particles to the VCC. This study identifies a new facet of how HIV interacts with macrophages, suggesting that disruption of this pathway could be a therapeutic strategy with implications for HIV cure efforts.

## Introduction

Tissue macrophages are an important target of HIV-1 infection. Infected macrophages have been found in numerous tissues derived from HIV-infected individuals, including peripheral lymph nodes, gut-associated lymphoid tissue, liver, lung, genitourinary tract, bone, and the brain [1–8]. Infected macrophages are thought to play a prominent role in HIV-related comorbidities, including HIV-associated neurocognitive disorders (HAND), accelerated cardiovascular disease, early immune aging, and endocrine disorders [9–13]. Infected macrophages may also form a long-lived reservoir for HIV-1 [14–16]. It is therefore important to understand how HIV-1 interacts with and alters macrophage structure and function.

HIV-1-infected macrophages form an unique intracellular structure known as the virus-containing compartment (VCC), also referred to as the intracellular plasma membrane compartment (IPMC) [17–20]. The VCC shares many molecular features of the late endosome/multivesicular body (MVB), including the presence of MHC class II, LAMP-1, and tetraspanins CD9, CD37, CD53, CD63, CD81, and CD82. The VCC is also enriched in β2 integrins, the scavenger receptor CD36, hyaluronate receptor/surface glycoprotein CD44, and phosphatidylinositol 4,5 bisphosphate (PIP$_2$) [17,21,22,22]. In contrast to the late endosome, the VCC has a near neutral pH [18]. Tubular connections extending from the VCC to the extracellular milieu have been observed by fluorescence microscopy, transmission EM, and ion abrasion scanning EM [17,20,22,23]. The origin and function of these tubules of the VCC are unknown, although it has been suggested they could represent an exit route for viruses from the VCC [23]. The VCC may serve as a virion storage site for HIV-1 that can promote trans-infection upon contact with target T cells or uninfected macrophages, similar to the role a similar "holding" compartment plays in dendritic cells (DCs) [24, 25]. The presence of budding forms observed by EM on VCC membranes, and the general paucity of particles seen assembling on the PM of macrophages, has led to the conclusion that the VCC is an intracellular assembly compartment and the major site of assembly in macrophages [17,18,26,27]. However, the genesis of an intracellular membranous compartment made up of both late endosomal and plasma membrane components remains uncertain.

Siglec-1 was identified as an important myeloid cell lectin that binds and captures HIV-1 particles through interactions with gangliosides on the lipid envelope of the virus, contributing to trans-infection of target cells by DCs [28, 29]. We have previously shown that Siglec-1 on the surface of monocyte-derived macrophages (MDMs) captures exogenous non-infectious

HIV-1 virus-like particles (VLPs) and leads to their internalization into the macrophage to form a structure identical to the VCC in morphology, location, and in the presence of typical VCC markers [30]. Furthermore, when HIV-1 VLPs are added exogenously to infected cells that contain a pre-formed VCC, they are delivered in a Siglec-1-dependent manner into the authentic VCC. Siglec-1 depletion inhibits VCC formation within infected MDMs and prevents a VCC-like structure from forming following exogenous addition of VLPs, suggesting a common role for Siglec-1 in VCC formation. Capture of HIV-1 on the surface of activated DCs leads to nanoclustering of Siglec-1, and this nanoclustering together with activation of the actin cytoskeleton is required for internalization to the VCC [31]. However, it remains unknown how the development of virus-Siglec-1 nanoclusters leads to particle uptake to the VCC.

Here we evaluated the role of cellular endocytic pathways in the uptake of HIV-1 particles following Siglec-1-mediated particle capture. Inhibition of dynamin-2 through chemical and genetic means resulted in the arrest of captured particles and Siglec-1 in a more peripheral location in MDMs, preventing VCC formation. Inhibition of clathrin-mediated endocytosis did not arrest VCC formation, leading us to focus on clathrin-independent pathways. Remarkably, clathrin-independent carriers/glycosylphosphatidylinositol-anchored protein-enriched endocytic compartments (CLIC/GEEC) cargo were highly enriched in the VCC of infected cells, and disruption of the CLIC/GEEC pathway prevented VLP uptake in infected cells and inhibited trans-infection of T cells. Furthermore, tubular membranes enriched in the CLIC/GEEC cargo protein CD98 were found to contain Siglec-1-captured VLPs. Migration of Siglec-1, CLIC/GEEC cargo proteins, and VLPs to the center of the cell over time resulted in the formation of structures identical to the authentic VCCs of HIV-1-infected macrophages. These results demonstrate that the CLIC/GEEC pathway is involved in the movement of Siglec-1-capture particles to the VCC and is required for VCC formation in infected MDMs.

## Results

### Dynamin inhibition or depletion prevents VLP uptake and VCC formation in MDMs

We first asked if dynamin is required for VLP uptake and VCC formation following particle capture by Siglec-1. The dynasore analog, Dyngo4a, has been shown to exhibit specific and potent inhibition of dynamin-mediated endocytosis [32, 33]. We therefore examined the effects of Dyngo4a treatment on HIV-1 VLP uptake into a perinuclear VCC in uninfected MDMs. MDMs were incubated with GFP-tagged HIV-1 VLPs for 5 hours to allow a VCC-like structure to form, after which they were treated with Dyngo4a or DMSO control for 30 minutes. mCherry-tagged VLPs were then added to the MDMs to assess the effects of dynamin inhibition on VLP uptake and delivery to the VCC. Transferrin was added 30 minutes prior to fixation to monitor dynamin-dependent uptake. As expected, Dyngo4a-treated cells were deficient in uptake of transferrin as compared with control cells (Fig 1A, 1B, transferrin panels, with quantitation in Fig 1D). HIV-1 mCherry VLP uptake was reduced but not eliminated by Dyngo4a treatment (Fig 1A, 1B, and volume measurement in Fig 1C). The VLPs that did enter the cell largely failed to reach the VCC following Dyngo4a treatment (Fig 1A, 1B, and colocalization data in Fig 1E), while in control cells mCherry VLPs colocalized extensively with GPF VLPs within the VCC. The HIV-1 mCherry VLPs in Dyngo4a-treated cells appeared to enter the cell, but were located in a position superior to the VCC, as if arrested in transit. Notably, Dyngo4a treatment specifically blocked dynamin-dependent endocytosis in MDMs, while not showing off-target effects on clathrin-independent endocytosis pathways, such as CLIC/GEEC and macropinocytosis, as indicated by preserved uptake of CD98 and low molecular

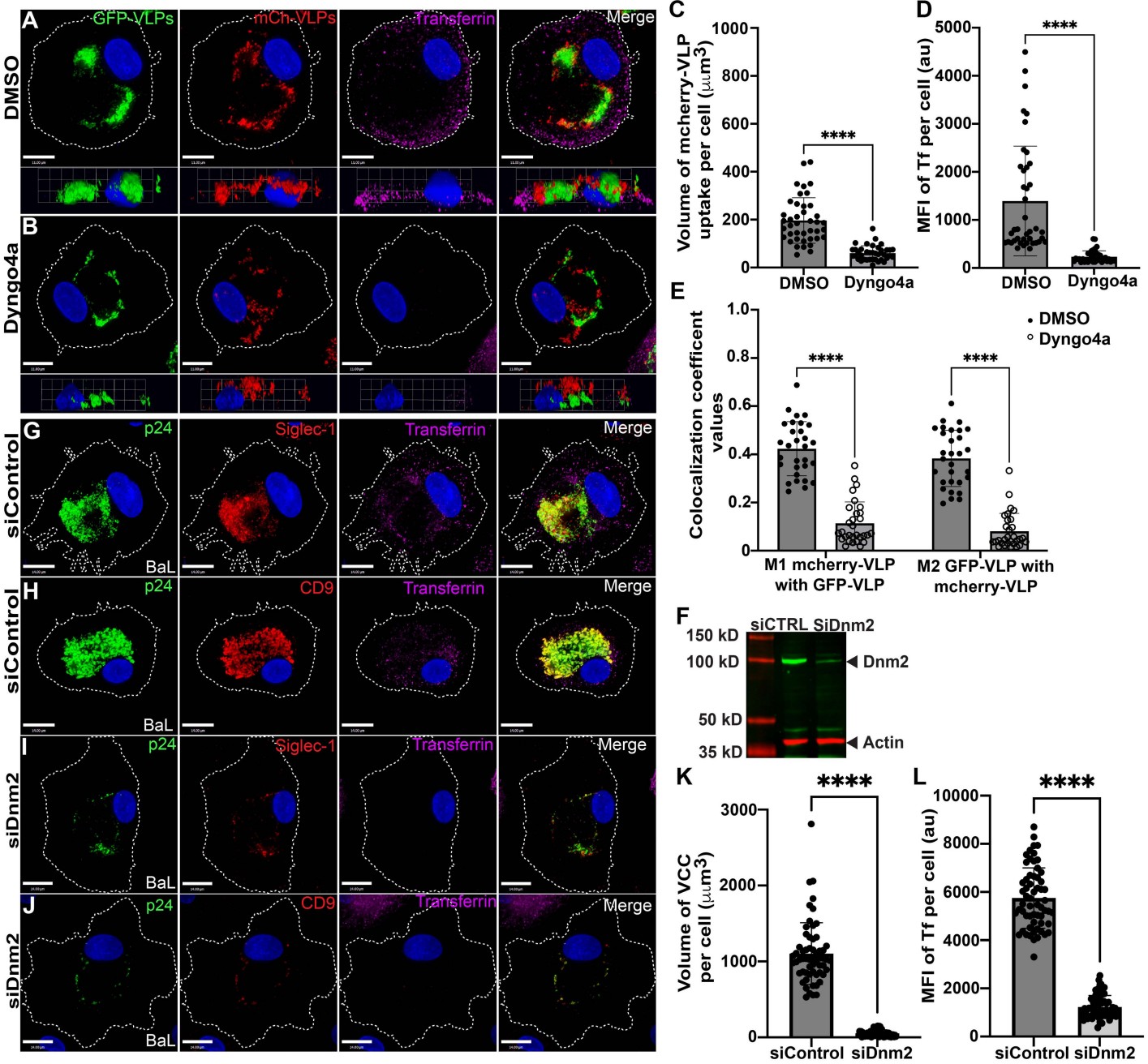

**Fig 1. Dynamin plays an important role in VLP uptake and VCC formation in MDMs.** HIV-1 Gag-GFP VLPs were added to MDM cultures and allowed to be internalized for 5 hours. MDMs were then treated with either DMSO (A) or Dyngo4a (B), followed by addition of HIV-1 Gag-mCherry VLPs for 2.5 hours (red), in order to assess uptake into the VCC. Transferrin uptake (magenta) was evaluated during the final 30 minutes before fixation. Size bar = 11 µm. (C) Mean ± SD volume of mCherry-VLP uptake per cell ($\mu$m$^3$) from a total of 30 MDMs in each group. (D) Mean ± SD fluorescence intensity of transferrin uptake per cell. (E) Colocalization coefficient values for mCherry VLP/GFP VLP (M1) and GFP VLP/mCherry VLP (M2). Open circles indicate Dyngo4a-treated cells. (F) Representative western blot showing depletion of Dnm2 in MDMs. (G,H) HIV-1$_{BaL}$-infected MDMs transfected with 50nM control siRNA, stained for p24 and VCC markers. (I,J) HIV-1$_{BaL}$-infected MDMs transfected with 50nM dynamin-2 siRNA. Transferrin uptake is shown in magenta, p24 staining in green, and Siglec-1 or CD9 in red. Scale bar = 14 µm. (K) Mean ± SD volume of the VCC per HIV-1$_{BaL}$-infected cell (60 different HIV-infected MDMs per group). (L) Mean ± SD fluorescence intensity of transferrin uptake per infected cell. Data are representative of a minimum of three independent experiments. ****p <0.001 and ns, not significant (Mann-Whitney test).

weight dextran (S1A-S1D Fig). These results suggest that HIV-1 virion uptake into the VCC is dynamin-dependent, while in the absence of dynamin activity VLPs are still captured, but are aborted in reaching the VCC.

To further evaluate the role of dynamin in particle uptake and VCC formation, we utilized siRNA-mediated depletion of dynamin-2. We focused on dynamin-2 as it is ubiquitously expressed in all cells, whereas dynamin-1 is expressed in primarily in neurons and dynamin-3 is expressed primarily in brain, lung and testis [34, 35]. MDMs were infected with HIV-1$_{BaL}$ and treated with control siRNA or dynamin-2 siRNA, and VCC formation evaluated on day 12 post-infection. Dynamin-2 expression levels were reduced by a mean of 78.8 ± 20.4% percent with dynamin-2 siRNA (Fig 1F, representative of 5 repeats with cells from different donors). The effectiveness of dynamin inhibition was also documented through addition of transferrin on day 12 post-infection, as uptake was inhibited in dynamin-2 siRNA-treated cells as compared with controls (Fig 1G-1J, with quantitation shown in Fig 1L). Control siRNA-treated MDMs formed deep VCCs, as indicated by colocalization of HIV-1 parti-cles, Siglec-1, and CD9 in a deep location in the cell (Fig 1G and 1H). Dynamin-2 depletion resulted in significant reduction in the VCC volume within HIV-1 infected MDMs (Fig 1I and 1J, with quantitation in Fig 1K). The mean VCC volume from BaL infection was reduced from 1104 ± 406 μm$^3$ to a mean of 53.5 ± 42.7 μm$^3$ (Fig 1K). 3D views showing the location of p24 along the z-axis for these same cells are shown in S1G-S1J Fig. To confirm that depletion of dynamin-2 did not prevent virus particle production in culture, thus indirectly reducing VCC formation, we measured p24 release over time following control or dynamin-2 siRNA treatment. Although there was some variation from experiment to experiment, particle pro-duction and release from infected MDMs following siRNA treatment overall was not inhibited by dynamin-2 depletion (S2 Fig). Together, chemical inhibition experiments and dynamin-2 depletion experiments support an important role for dynamin-2 in the internalization of Siglec-1-captured particles and in VCC formation.

## Clathrin-mediated endocytosis (CME) is not required for Siglec-1-mediated particle uptake and VCC formation in macrophages

Endocytosis through CME requires dynamin for the scission and release of clathrin-coated vesicles from the plasma membrane [36, 37]. To evaluate the potential role of CME in VLP uptake, we initially performed chemical inhibition experiments using Pitstop 2.0. However, this inhibitor demonstrated significant off-target effects in MDMs, including disruption of clathrin-independent endocytosis (CIE) pathways (S1E and S1F Fig) as has been previously published, limiting its utility for our studies [38]. Therefore, we relied on depletion of key components of the CME pathway, epidermal growth factor receptor substrate 15 (Eps15) and Fer/CIPa homology domain protein 2 (FCHO2), both of which contribute to early steps of the CME pathway [36,37,39]. Eps15 expression levels in MDMs were reduced by a mean of 73.5 ± 19% following siRNA-mediated depletion, and FCHO2 depletion was 69.5 ± 17% in repeated experiments (representative blots in Fig 2B and 2C). On day 6 following siRNA treatment, HIV-1 VLPs were added and allowed to be internalized for 15 hours. Transferrin was added 30 minutes prior to fixation to monitor CME-mediated uptake. Control siRNA-treated MDMs demonstrated robust uptake of transferrin (Fig 2A, with quantitation in Fig 2G). As expected, depletion of either Eps15 and FCHO2 resulted in a significant reduction of transferrin uptake (Fig 2D and 2E, with quantitation in Fig 2G). Despite robust inhibition of CME, there was no significant difference in the volume of the compartment measured by VLP uptake in MDMs depleted of Eps15 or FCHO2 vs. control MDMs (Fig 2D, 2E, with quantitation in Fig 2F). 3D views showing the location of VLPs along the z-axis for these

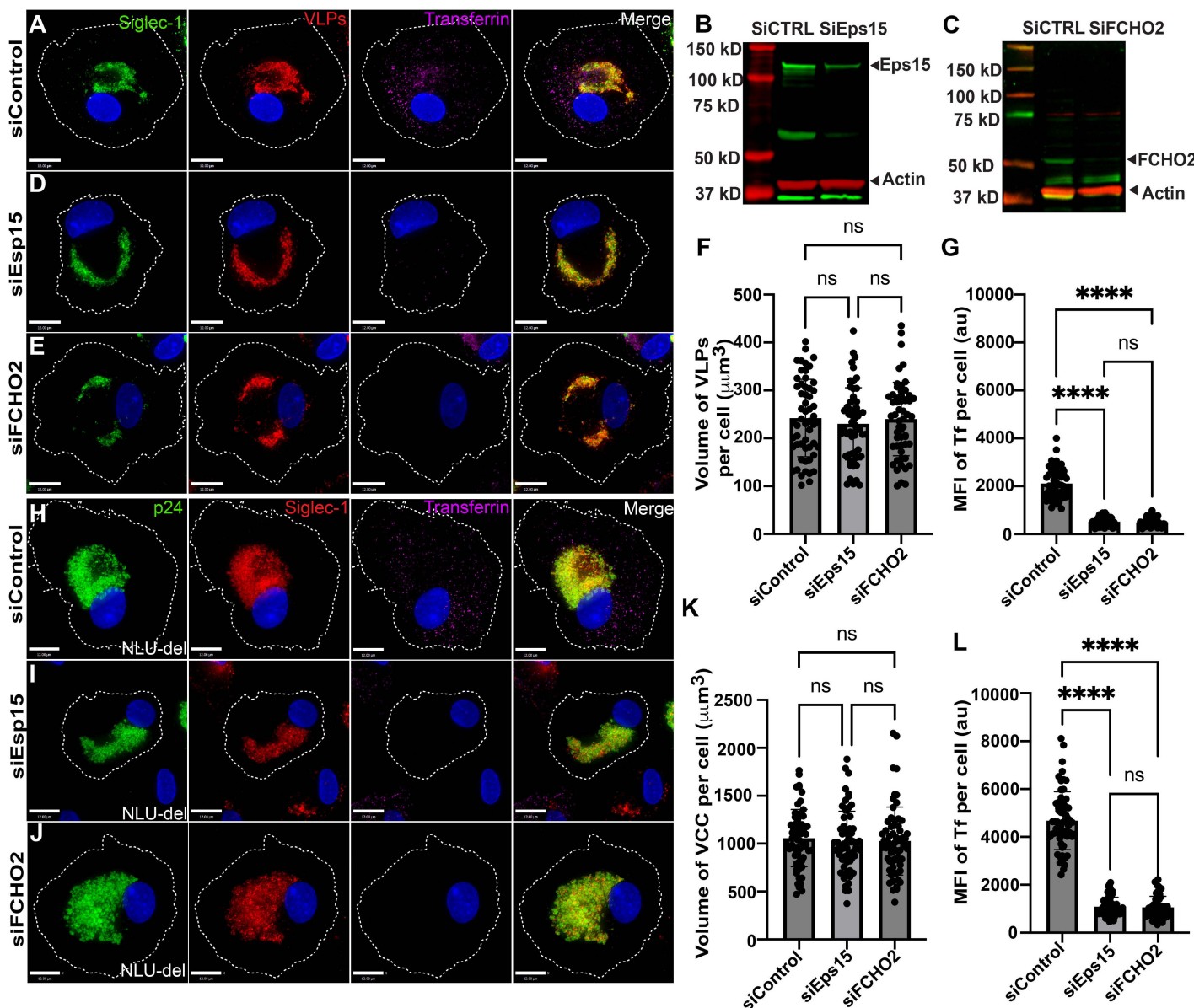

**Fig 2. Inhibition of CME does not prevent formation of the VCC in MDMs.** (A) MDMs transfected with 50nM of control siRNA, showing uptake of mCherry VLPs (red) and transferrin (magenta) together with Siglec-1 (green). (B) Representative knockdown of Eps15 in MDMs transfected with siControl or siEps15. (C) Representative knockdown of FCHO2 in MDMs transfected with siControl or siFCHO2. (D) MDMs transfected with siEps15 and stained as described for (A). (E) MDMs transfected with siFCHO2 and stained as described for (A). (F) Mean ± SD volume of VLPs per cell from 56 different MDMs for each experimental group. (G) Mean ± SD fluorescence intensity of transferrin per cell. (H) NLUdel-infected MDMs transfected with control siRNA, showing p24 (green), Siglec-1 (red), and transferrin (magenta). Scale bar = 12 μm. (I) NLUdel- infected MDMs treated with Eps15 siRNA and stained as before. (J) NLUdel-infected MDMs treated with FCHO2 siRNA. Scale bar = 12 μm. (K) Mean ± SD volume of VCC per cell from a total of 60 different NLUdel-infected MDMs in each group. (L) Mean ± SD fluorescence intensity of transferrin uptake per cell. Data are representative of a minimum of three independent experiments. ****p <0.001 and ns, not significant (Kruskal-Wallis test followed by Dunnett's multiple comparisons test).

same cells are shown in S3A-S3C Fig. To further assess the role of the CME pathway in VCC formation, we performed depletion of Eps15 and FCHO2 in HIV-1-infected MDMs. We first infected the MDMs overnight with VSV-G-pseudotyped *vpu*-deficient NLUdel, then utilized siRNA against Eps15, FCHO2 or control as before. Note that NLUdel is a *vpu*-deficient virus has previously been used to study VCC formation, producing enlarged VCCs due to lack of

downregulation of tetherin [30]. Control siRNA-treated MDMs formed VCCs deep within the cell body and demonstrated robust transferrin uptake (Fig 2H, with quantitation in Fig 2L). Depletion of Eps15 or FCHO2 led to inhibition of transferrin uptake as expected (Fig 2I and 2J and quantitation shown in Fig 2L). No significant change in VCC volume was seen following depletion of either component of the CME pathway (Fig 2I and 2J, quantitation in Fig 2K). VCCs were located deep within the cell, as shown in 3D views of the cells from Fig 2 panels H-J, shown in S3D-S3F Fig. These data suggest that the CME pathway is not involved in the formation of the VCC in HIV-1-infected MDMs. Based on these results, we turned our attention to clathrin-independent pathways of endocytosis.

We considered a potential role for caveolin in HIV-1 particle uptake in MDMs. The expression of caveolins in murine macrophages along with partitioning in lipid rafts has been established, although data for expression in human macrophages has been somewhat conflicting [40]. Some reports have demonstrated that human MDMs fail to express caveolin 1, 2, or 3 [41]. We first looked for CAV1, CAV2, and CAV3 expression using RNAseq data from MDMs that we had recently infected with the same HIV-1$_{BaL}$ strain [42]. The level of expression of all three caveolins at the RNA level was extremely low or undetectable, including following GM-CSF maturation and type I IFN stimulation to enhance Siglec-1 expression (as used in other experiments in this study), and also following infection with HIV-1 (S1 Table). Housekeeping genes GAPDH and beta actin are shown for comparison. Protein expression was next evaluated. Caveolin expression in HeLa cells was readily detected by western blotting (S4 Fig). However, caveolin 1, 2, and 3 were undetectable in MDM cell lysates (representative blots are shown in S4 Fig). We conclude that caveolin is not expressed at detectable levels in the human MDMs prepared for the experiments here, while VCC formation is prominent, leading us to examine other CIE pathways that may play a role in particle internalization and VCC formation.

### The VCC in HIV-1-infected human MDMs is enriched in CLIC/GEEC cargo CD44 and CD98

The hyaluronate receptor and surface glycoprotein CD44 has previously been described as a component of the VCC in infected MDMs [22]. Because CD44 is a characteristic cargo protein of the CLIC/GEEC endocytosis pathway [43, 44], we hypothesized that the CLIC/GEEC pathway may play a role in VCC formation. We examined the subcellular distribution of a second characteristic CLIC/GEEC cargo protein, the amino acid transporter CD98. MDMs were infected with HIV-1$_{BaL}$ and incubated for 10 days to allow VCC formation. Established markers of the VCC, CD9 and Siglec-1, colocalized with virus in the VCC as expected (Fig 3A and 3B). CD44 was highly concentrated in the VCC as previously described (Fig 3C). Notably, CD98 was also highly enriched in the VCC, identifying a second classic marker of the CLIC/GEEC pathway in this compartment (Fig 3D). The cellular location of the VCC in these experiments was deep within the cell as expected for the VCC, with 3D views corresponding to the X-Y views of Fig 3A-3D included in S5A-S5D Fig. Colocalization between CD98 and p24 in these experiments was extensive, and is quantified in S5E Fig. These findings indicating enrichment of CLIC/GEEC cargo in the VCC then led us to further examine the potential functional role of this pathway in the formation of the VCC.

### Chemical inhibition of the CLIC/GEEC pathway inhibits VLP uptake into MDMs

We next examined the effect of chemical inhibition of the CLIC/GEEC pathway using 7-keto-cholesterol (7-KC). 7-KC is an oxysterol that prevents close packing of acyl chains and

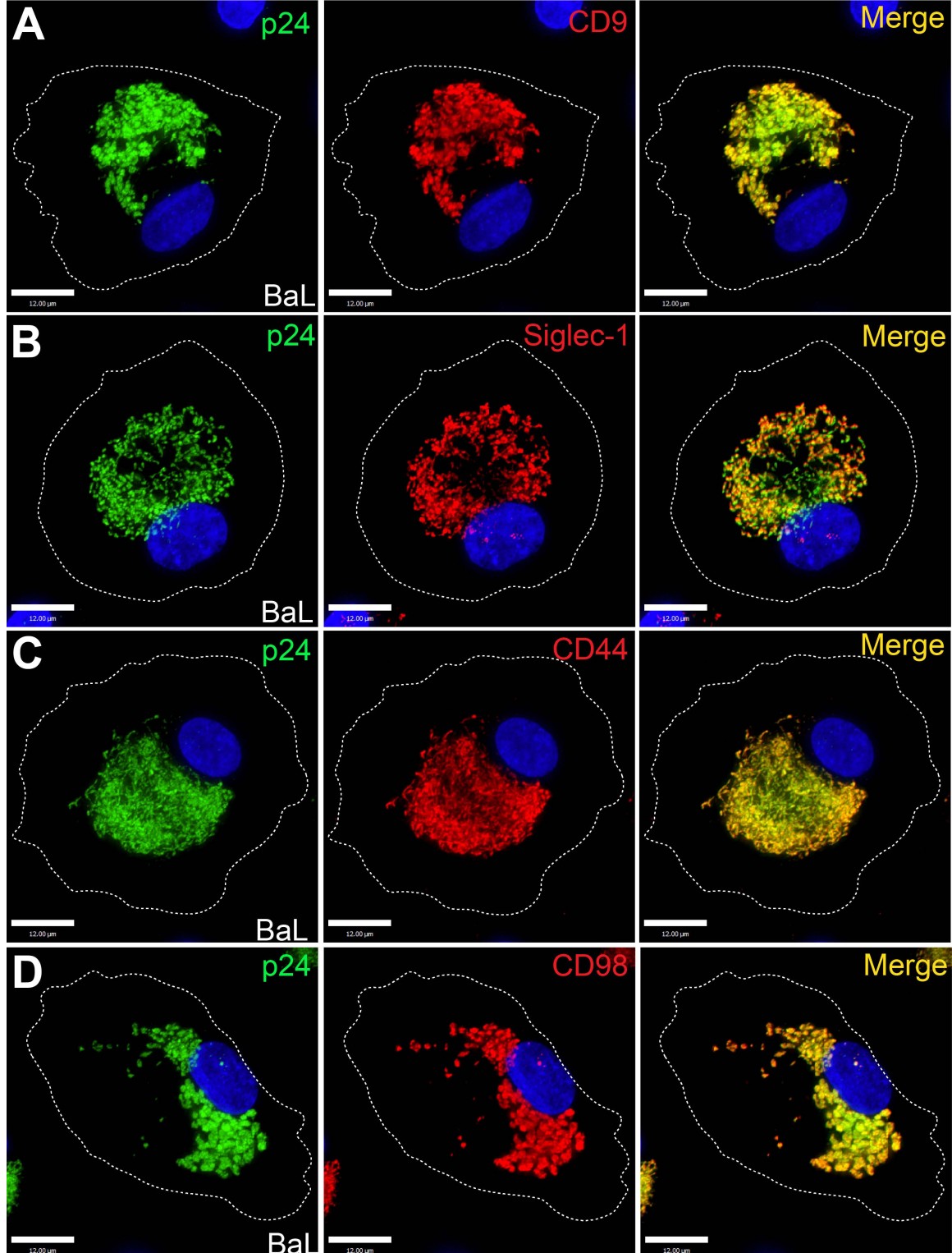

**Fig 3. CLIC/GEEC Cargo in the VCC in HIV-1 BaL-infected MDMs.** On day 12 post-infection, MDMs were fixed and immunos-tained for p24 (green) and (A) CD9, (B) Siglec-1, (C) CD44, or (D) CD98 (red). Scale bar = 12 μm. Results shown are representative of repeated experiments performed using MDMs derived from 3 different donors.

has been shown to inhibit endocytosis via the CLIC/GEEC pathway [45, 46]. MDMs were pre-treated with 7-KC prior to the addition of fluorescent HIV-1 VLPs, and the internalization of VLPs compared with that seen in control cells. To monitor CLIC/GEEC inhibition, we utilized a pulse-chase with anti-CD44 or anti-CD98 antibodies, applied 30 minutes prior to fixation. Control MDMs readily took up VLPs into VCCs, where they colocalized with Siglec-1, CD98, and CD44 (Fig 4A and 4B, quantitation in Fig 4E, F and I). Treatment with 7-KC reduced the amount of CD98 antibody and CD44 antibody internalized, and also markedly inhibited VLP uptake. After treatment with 7-KC, VLPs captured by Siglec-1 were prominent on the cell surface (Fig 4C-4D, see 3D side views). Data with 7-KC treatment thus supports the hypothesis that the CLIC/GEEC pathway is required for the uptake of VLPs leading to formation of the VCC.

To further evaluate the involvement CLIC/GEEC pathway in the formation of the VCC, we examined the effects of 7-KC treatment on uptake of VLPs into a pre-formed VCC in HIV-1-infected MDMs. MDMs were first infected with VSV-G-pseudotyped *vpu*-deficient NLUdel, allowing the formation of enlarged VCCs. On day 10 post-infection, MDMs were treated with 7-KC, followed by VLP addition to evaluate endocytosis to the pre-formed VCC. Note that we used KC57 antibody to detect mature p24 produced following infection (green), allowing us to distinguish signal from the uncleaved Gag core of mCherry-VLPs (red), as had been previously established [30]. Control MDMs demonstrated robust VLP uptake and were concentrated within the VCC of infected MDMs (Fig 4G, quantitation in Fig 4I). In contrast, 7-KC-treated MDMs failed to endocytose VLPs into the pre-formed VCC (Fig 4H, quantitation in Fig 4I). Notably, as had been seen previously with 7-KC treatment followed by VLPs addition, the added VLPs were captured by Siglec-1 but remained predominantly on the surface of the MDMs (Fig 4H, compare to Fig 4G). Thus, experiments using 7-KC indicate that this inhibitor of CLIC/GEEC endocytosis prevents uptake of Siglec-1-captured particles to the VCC.

To address the potential that off-target effects of 7-KC could be responsible for the observed inhibition of particle uptake, we confirmed that while 7-KC prevented internalization of CD44 and CD98, it did not inhibit CME, as shown by preserved transferrin uptake in the presence of inhibitor (S6A-S6D Fig, with quantitation in S6E-S6G Fig). Furthermore, 7-KC treatment of MDMs did not inhibit phagocytosis of IgG-opsonized latex beads (S6H-S6J Fig), and did not diminish the viability of MDMs in a 12-hour exposure (S6L Fig). We conclude that 7-KC did not interfere with endocytosis via the CME pathway or with phagocytosis, and did not diminish cell viability in the time frame utilized in our experiments.

## Inhibition of CLIC/GEEC-mediated endocytosis by IRSp53 depletion prevents particle uptake and VCC formation

Insulin-responsive protein of mass 53kDa (IRSp53) is a BAR domain protein known to interact with CDC42 and ARF1, mediating membrane curvature and interacting with actin regulatory proteins necessary for CLIC/GEEC endocytosis [47, 48]. We next examined depletion of IRSp53 as a second method of examining the role of the CLIC/GEEC pathway in VCC formation. Control siRNA-treated MDMs demonstrated robust uptake of VLPs, forming the VCC as expected, and IRSp53 staining was readily detected (Fig 5A). Depletion of IRSp53 was performed using siRNA, and was evident by immunostaining (Fig 5B) and by western blotting (Fig 5C). The extent of knockdown was 73.2 ± 6.1% as judged by western blot from 3 separate experiments. Depletion of IRSp53 resulted in marked reduction in the volume of internalized VLPs, with only a few scattered VLPs visible within the cell (Fig 5B, quantitation in Fig 5D). These findings support a model in which inhibition of the CLIC/GEEC pathway inhibits internalization of Siglec-1-captured VLPs and eliminates VCC formation.

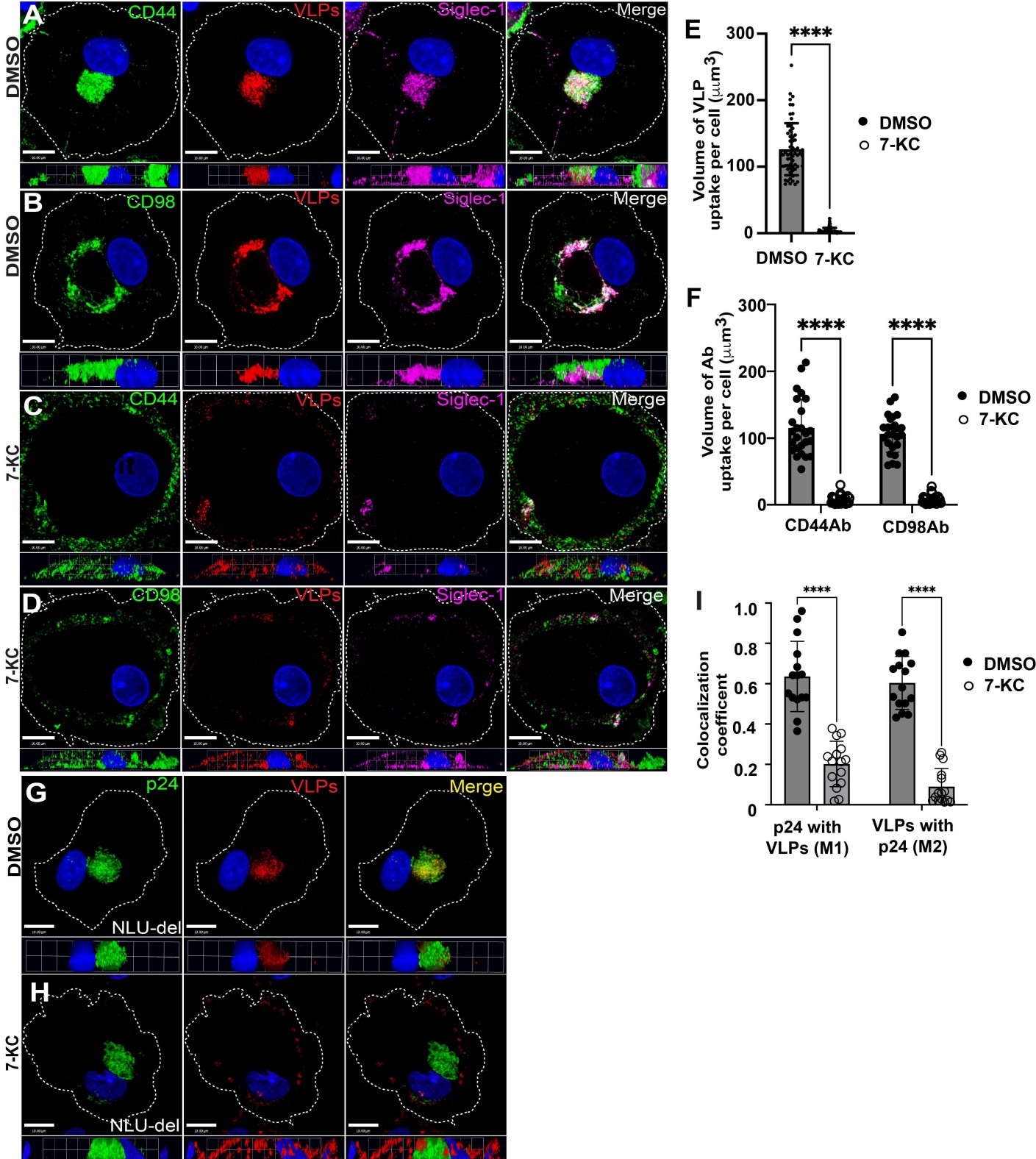

**Fig 4. 7.KC treatment prevents VLP internalization to the VCC.** (A and B) MDMs were treated with DMSO control and incubated for 12 hours, followed by addition of mCherry HIV-1 VLPs for 14 hours to allow VLP uptake. Pulse-chase with CD44 Ab (green), Siglec-1 Ab (magenta), and CD98 Ab (green) for 30 minutes was performed prior to fixation of MDMs. Scale bars = 10μm. (C and D) MDMs were treated with 7-KC for 12 hours, followed by VLP addition and pulse-chase

with antibodies against CD44, CD98, and Siglec-1 as in (A, B). Images shown are representative maximal Intensity projections of deconvolved z-stacks from the MDMs, along with 3D side projection from the same images below. (E) Volume of VLP per cell ($\mu m^3$) of control vs 7-KC treated MDMs, mean values ± SD (****P value of <0.0001) from 60 cells/group. (F) Volume of CD44 and CD98 uptake per cell for control and 7-KC-treated cells (mean ± SD, ****P value of <0.0001). (G) MDMs were infected with NLUdel, followed by addition of mCherry VLPs on day 10 post-infection to evaluate the internalization of VLPs (red) to the VCC (green, p24). Note that VLPs reach the pre-formed VCC where they colocalize with mature virions. Scale bar = 10μm. (H) MDMs were infected by NLUdel as in (G), then treated with 7-KC on day 10 post-infection at the time of mCherry-VLP addition. Note lack of colocalization of VLP signal with pre-formed VCC (green). (I) Colocalization coefficient (M1) of p24 with mCherry VLPs and (M2) of mCherry VLPs with p24 in control vs. 7-KC-treated cells. Values represent measurements from 15 cells/group. ****p <0.001 and ns, not significant (Mann-Whitney test). Experiments were performed with MDMs derived from three different donors.

We further examined the role of IRSp53 on VCC formation within HIV-1$_{BaL}$ -infected MDMs by depleting IRSp53 beginning one day post-infection, incubating cells for 12 days, and staining for p24 and VCC markers. Control siRNA-treated MDMs formed large deep VCCs as measured by p24, Siglec-1, and CD9 staining (Fig 5E and 5F). IRSp53-specific siRNA treatment of MDMs resulted in a reduction of IRSP53 staining of 70% as measured by fluorescence intensity (Fig 5J). IRSp53 depletion resulted in a significant reduction in VCC volume within HIV-1 infected MDMs (Fig 5G and 5H), with mean VCC volume reduction from 822 ± 265 $\mu m^3$ to 17 ± 16 $\mu m^3$ (Fig 5I). The observed reduction in VCC volume was not due to an inhibition of particle formation following depletion of IRSp53, as indicated by measurement of p24 release from infected MDM supernatants (S2 Fig). We conclude that both chemical inhibition using 7-KC and genetic inhibition through depletion of IRSp53 inhibits uptake of HIV-1 particles following capture by Siglec-1, supporting a role for the CLIC/GEEC pathway in VCC formation within HIV-1-infected MDMs.

## Role of the CLIC/GEEC pathway and dynamin-2 in trans-infection from infected MDMs to T-lymphocytes

The VCC in DCs and MDMs is thought to play a role in HIV spread within infected individuals through trans-infection of susceptible target cells upon cell-cell contact [30,49]. We therefore evaluated the role of dynamin-2 and the CLIC/GEEC pathway on transmission of virus from infected MDMs to CD4+ T cells. MDMs were infected with HIV-1$_{BaL}$, followed by depletion of IRSp53 or dynamin-2. On day 7 post-infection, MDMs were treated with indinavir or control media prior to the addition of autologous T cells. Indinavir was employed to render newly-formed viruses non-infectious during the period of cell-cell contact, therefore limiting trans-infection solely to pre-formed viruses from the population within the VCC. Following a 12-hour period of contact, T-cells were separated from MDMs, cultured for an additional 24 hours in the presence of protease inhibitor to prevent further spread of infection, then stained for surface CD3 and intracellular p24 antigen for analysis by flow cytometry. The experimental schema is depicted in Fig 6A, with representative siRNA depletion and gating strategy provided in S7A-S7D Fig. Remarkably, depletion of either IRSp53 or dynamin-2 resulted in significantly diminished trans-infection of T cells following co-culture (Fig 6B and 6C). In the absence of indinavir, infection percentage was reduced from 6.8 ± 3.1% to a mean of 3.3 ± 1.2% following dynamin-2 depletion, and to 1.9 ± 1.1% following IRSp53 depletion, representing transmission of both newly-formed virus and virus from the VCC (Fig 6B and 6C). The efficiency of trans-infection was lower in the presence of indinavir, but was still significant, and was reduced from 3.5 ± 1.5% to 1.4 ± 0.8% by dynamin-2 depletion and to 1.1 ± 0.6% by IRSp53 depletion, representing effects on transmission of virus originating from the VCC. Control experiments confirmed that indinavir given at early times post-infection of MDMs (day 3) was highly effective at inhibiting trans-infection due to its effect in preventing

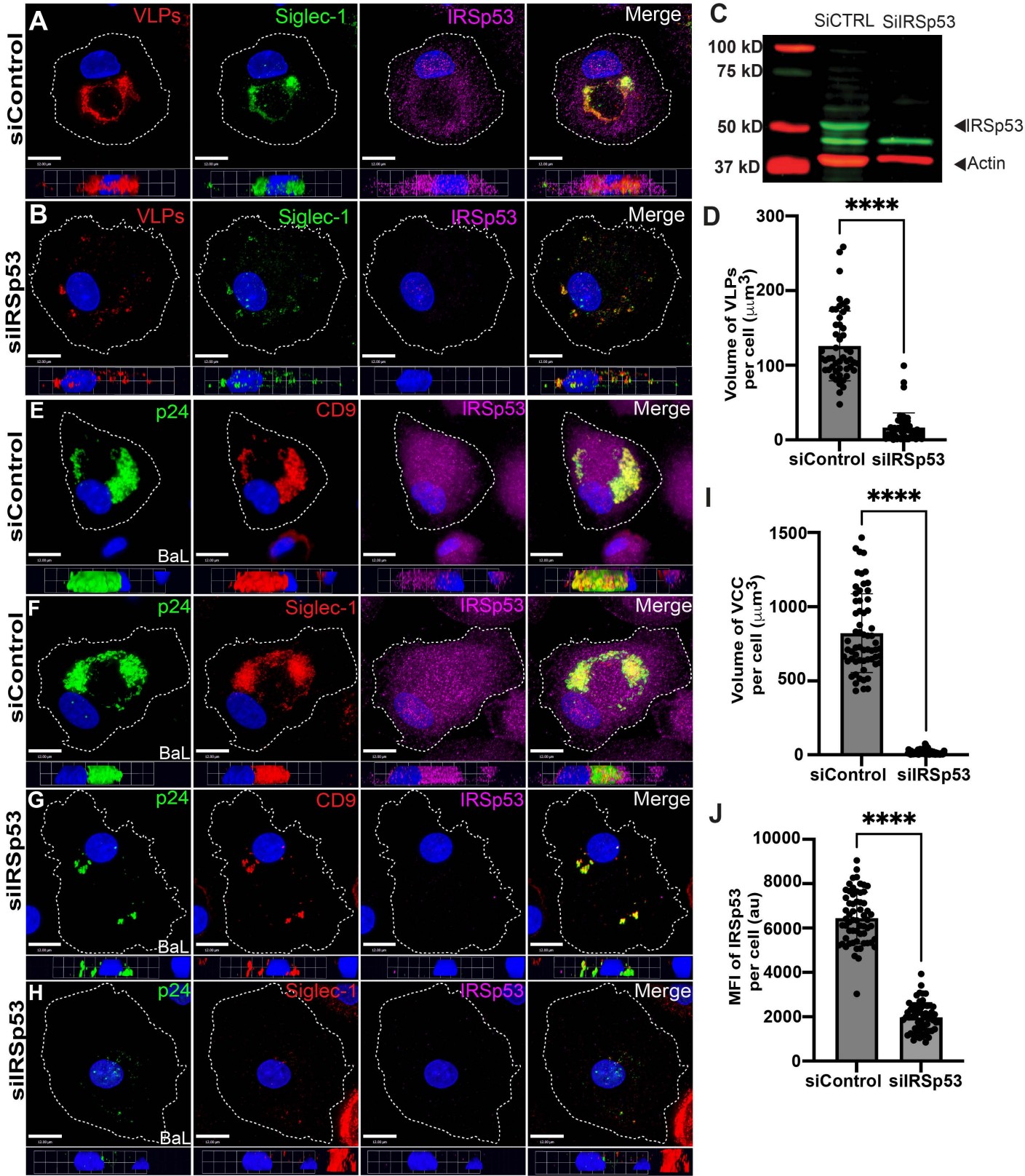

**Fig 5. IRSp53 depletion inhibits HIV-1 particle uptake and VCC formation.** (A) VLP uptake and VCC formation in MDMs transfected with 50nM of control siRNA, showing VCC formation represented by VLP and Siglec-1 staining; also showing IRSp53 staining. (B) VLP uptake in MDMs depleted of IRSp53 using

siRNA, with same markers as in (A). Scale bar = 12 μm. (C) Representative knockdown of IRSp53 in MDMs that were transfected with either siControl or siIRSp53, as shown by western blot. (D) Mean ± SD volume of VLP volume per cell, data from a total of 50 MDMs per group. (E and F) HIV-1$_{BaL}$- infected MDMs were transfected with 50nM control siRNA on day one following infection, then stained for the indicated markers on day 12. Scale bar = 12 μm. (G and H) HIV-1$_{BaL}$-infected MDMs transfected with siIRSp53 and stained for the indicated markers on day 12 post-infection. Note depletion of IRSp53 and reduced size of VCC markers. (I) Mean ± SD volume of VCC per cell; data from a total of 61 BaL-infected MDMs in each group. (J) Mean ± SD measurements of the MFI of IRSp53 expression. Data are representative of at least three independent experiments ****p <0.001 and ns, not significant (Mann-Whitney test).

the formation of a VCC (S7E Fig). These results provide an important functional correlate of the role of both dynamin-2 and the CLIC/GEEC pathway in VCC formation, extending the importance of both pathways to HIV-1 trans-infection from infected MDMs to target T cells.

## Siglec-1-captured VLPs are internalized within membrane tubules together with CLIC/GEEC cargo

Membranous tubules have been shown to emanate from the VCC of infected macrophages [17,50]. The presence of HIV-1 particles within 150–200 nm membranous tubules connecting to the PM has been documented by ion abrasion scanning electron microscopy [23]. When examining uninfected MDMs, we observed CLIC/GEEC tubules (as indicated by CD98 staining) that extended from the cell surface toward the center of the cell (Fig 7A). Siglec-1 was found primarily in small puncta throughout the cell and on the cell surface, with some limited colocalization with CD98-marked tubules in these uninfected cells (Fig 7A). We hypothesized that the virion-containing tubules previously identified connecting the PM to the VCC may represent Siglec-1-captured virus from the PM within tubules of the CLIC/GEEC pathway. To evaluate this possibility, we performed timecourse experiments in which we pulse-labeled CD98 to mark CLIC/GEEC tubules and then monitored the movement of Siglec-1-captured particles and CD98 from the surface of MDMs into the cell. Remarkably, we found the Siglec-1-captured VLPs concentrated along the CD98+ tubules at early time points following VLP addition (Fig 7C, 30 minutes). By 2 hours, the pulsed CD98 marker, together with Siglec-1 and VLPs, formed a more centralized coalescence of membranes in the perinuclear region, creating a VCC-like structure (Fig 7D). By 6 hours post-VLP addition, further coalescence of VLPs, CD98, and Siglec-1 had occurred, and a characteristic horseshoe-like VCC formed (Fig 7E). A 3D view of the 6 hour structure/VCC is shown in S8A Fig. Colocalization between VLPs and CD98 was apparent as early as 30 minutes following addition, and increased over time (quantified in S8B Fig). The presence of VLPs in CD98+ tubules (CLICs) and the coalescence of tubules together with Siglec-1 and p24 into a structure deep within the cell strongly suggests that the CLIC/GEEC pathway contributes to the genesis of the VCC.

## Inhibition of the CLIC/GEEC pathway using 7-KC prevents CLIC tubule formation and internalization of HIV-1 VLPs

The centripetal movement of CLIC/GEEC cargo CD98 together with VLPs shown above suggests that formation of CLIC tubules is an important step in VCC formation. We next examined the distribution of CLIC tubule marker CD98, VLPs, and Siglec-1 by labeling of CD98 at a timepoint 6 hours following VCC formation. Fig 8A shows that following labeling with anti-CD98 antibody, CD98 tubules are found in the center of the cell together with VLPs as before. In contrast, 7-KC treatment led to complete disruption of the tubular architecture marked by CD98 staining, and CD98 was seen largely on the cell surface (Fig 8B, with quantitation in Fig 8C and 8D). VLPs were captured by Siglec-1 under this condition, but remained

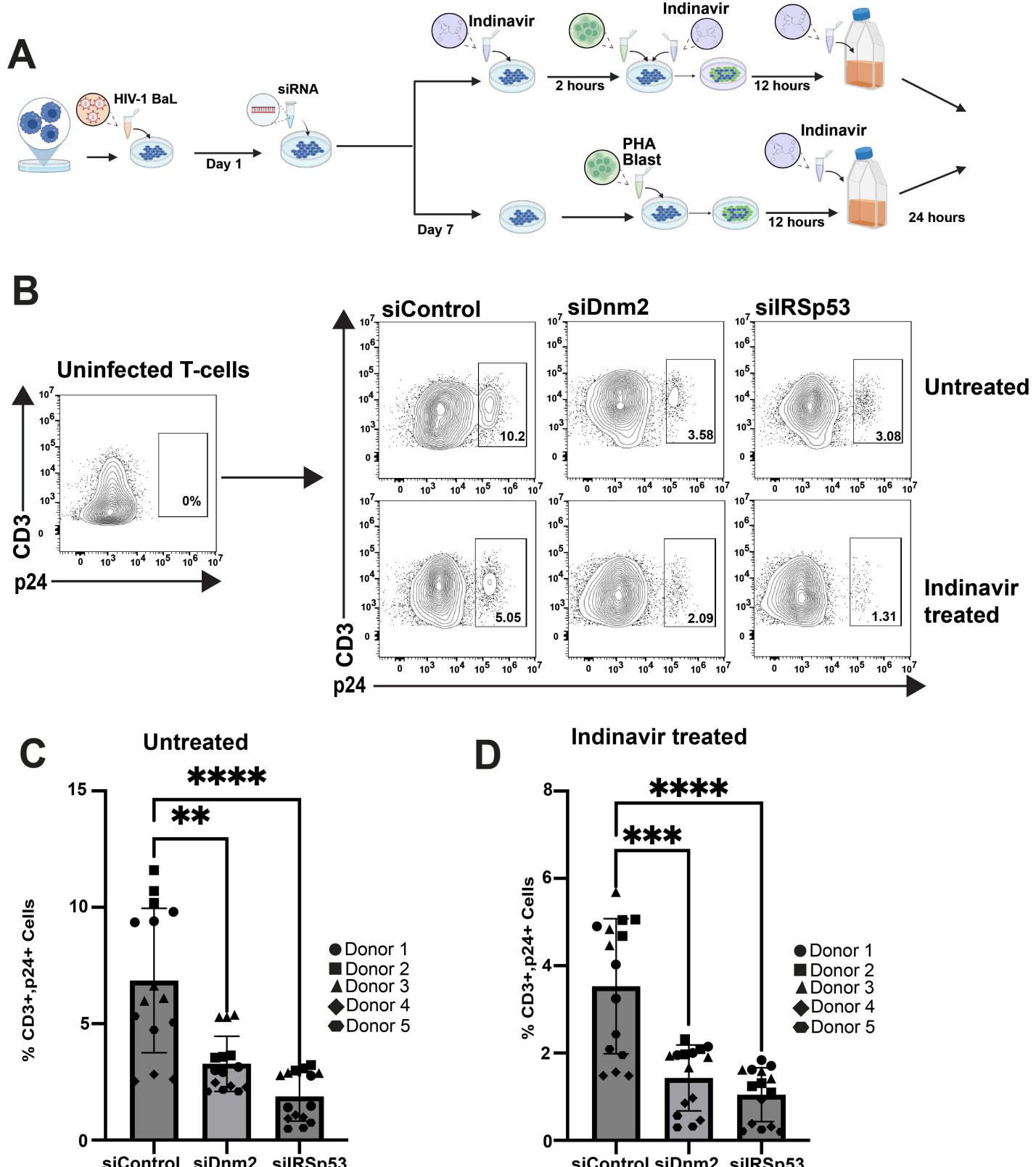

**Fig 6. Inhibition of the CLIC/GEEC pathway and dynamin-2 in MDMs reduces trans-infection of autologous T-cells.** (A) Schematic outline of the trans-infection experiment. MDMs were infected with HIV-1$_{BaL}$ (day 0). On day 1 post-infection, MDMs were transfected with either siControl, siIRSp53, or siDnm2. On

day 7 post-infection, MDMs were treated with either 10μM indinavir or control media for 2 hours prior to the addition of autologous PHA-stimulated T cell blasts. MDMs were cocultured with T cells for 12 hours, after which T cells were separated from MDMs and cultured for an additional 24 hours in the presence of indinavir (preventing spread from T cell to T cell). Flow cytometry of CD3+ p24+ stained cells was used to measure the efficiency of virus transmission to autologous T cells. (B) Flow cytometry plots gated for CD3+ and p24+ from a representative donor, showing the percent of HIV-1 transmission to T-cells. (C) Graph of the mean ± SD of the percentage of CD3+ and p24+ cells from 5 donors (3 technical replicates each). (D) Graph of the mean ± SD of the percentage of CD3+ and p24+ cells treated with Indinavir; data from 5 donors with 3 technical replicates each. **p=0.0021, ***p=0.0002, ****p <0.0001 and ns, not significant (Brown-Forsyth and Welch ANOVA test followed by Dunnett's multiple comparisons test). Schematic in (A) created in https://BioRender.com.

largely on the surface of the cell (Fig 8B, VLP and Siglec-1 panels). Together, these findings suggest that the formation of CLIC tubules is an essential step in the internalization of Siglec-1-captured HIV particles leading to VCC formation.

We previously showed that when dynamin-2 is depleted in infected MDMs, the VCC volume is reduced (Fig 1). We next examined the effect of dynamin-2 depletion on Siglec-1-captured VLP uptake and CLIC tubules, using CD98 staining in MDMs. Transferrin was added 30 minutes prior to fixation to monitor dynamin-dependent uptake. As expected, control siRNA-treated MDMs demonstrated robust uptake of VLPs and transferrin, with a mean VCC volume of 931 ± 387μm³ (Fig 8E and 8F, with quantitation in Fig 8I and 8J). Siglec-1-captured VLPs were strongly colocalized with CD98 in deep structures as previously observed (colocalization shown in Fig 8K and 8L). Dynamin-2 depleted MDMs were deficient in uptake of transferrin as compared with control (Fig 8E-8H, transferrin panels, with quantitation in 8J). Dynamin-2 depletion also resulted in a substantial reduction in the volume of internalized VLPs (mean of 106 ± 52μm³, Fig 8G and 8H, with quantitation in Fig 8I). In the dynamin-2 depleted MDMs, Siglec-1-captured VLPs were present in small CD98+ tubules in a more peripheral location (Fig 8G and 8H, with 3D view in S9 Fig). These results suggest that in the absence of dynamin-2, CLIC tubules bearing Siglec-1-captured VLPs are initiated, but are arrested in progressing to VCC formation.

## Discussion

Siglec-1 on the surface of macrophages or dendritic cells (DCs) captures infectious HIV-1 virions or HIV-1 VLPs by attaching to gangliosides on the lipid envelope of the virus, resulting in internalization of particles into the MDMs to form the VCC [28–31]. The compartment formed following capture of virions by Siglec-1 can facilitate trans-infection of T cells [28, 29]. Siglec-1 nanoclustering occurs in activated DCs, facilitating particle capture and leading to actin rearrangements that promote VCC formation, and this model likely applies to macrophages as well [31]. However, the exact mechanisms underlying viral particle uptake following capture by Siglec-1 remain unknown. Here we investigated candidate endocytic pathways involved in the internalization of HIV-1 particles for VCC formation in macrophages. We found that following Siglec-1-mediated capture, uptake of particles occurs via the CLIC/GEEC pathway. In contrast to classical CLIC/GEEC endocytosis, however, particle uptake into the VCC also requires dynamin-2 (Fig 9).

We found strong evidence for the involvement of the CLIC/GEEC pathway in HIV-1 particle uptake and VCC formation. It has been previously described that CD44, a classical marker for the CLIC/GEEC pathway, is enriched in the VCC [22,47,51]. We show that a second classical marker of the CLIC/GEEC pathway, CD98, is also highly concentrated in the VCC of the HIV-1 infected MDMs. Chemical inhibition or depletion of essential components of the CLIC/GEEC pathway diminished particle uptake and drastically reduced VCC volume, supporting a central role for the CLIC/GEEC pathway in VCC formation. Remarkably,

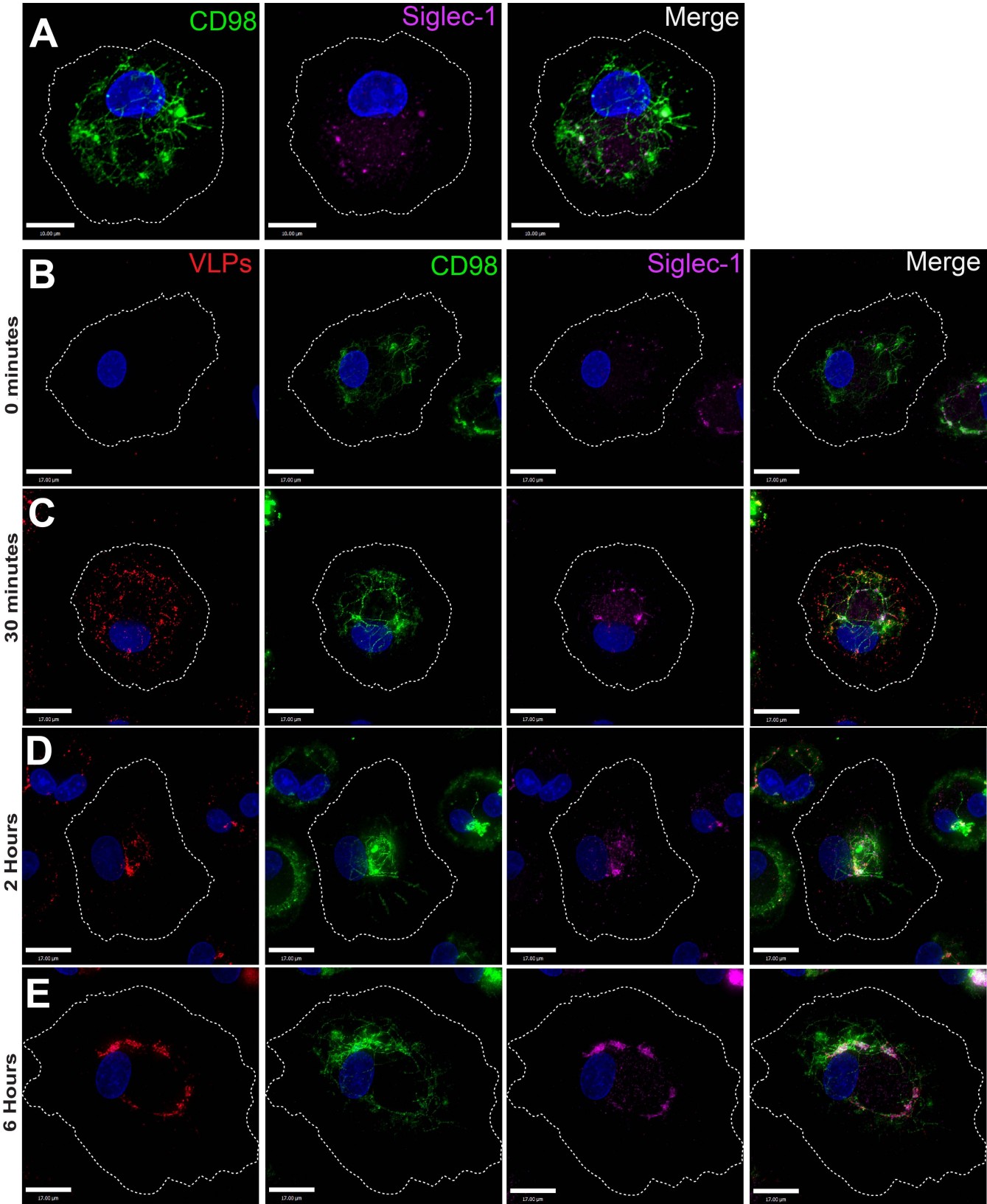

**Fig 7. Siglec-1-captured VLPs associate with CLIC/GEEC tubules and move centripetally to form a VCC-like structure.** (A) CD98Ab was added to MDMs for 1.5 hours before fixation, then cells were fixed and stained for Siglec-1 (magenta) and CD98 (green). Scale bar = 10 μm. (B-E) MDMs were pulsed

with CD98Ab (green) for 1.5 hours, then mCherry-VLPs (red) were added to the media followed by incubation for 0 minutes (B), 30 minutes (C), 2 hours (D), and 6 hours (E). At the indicated times, MDMs were washed, fixed in warm 4% PFA and secondary antibodies applied for immunostaining of Siglec-1 (purple) and CD98 (green). Images are representative maximal intensity projections derived from deconvolved z-stacks from MDMs. Data are representative of least three independent experiments. Scale bar = 17 μm for B-E.

Siglec-1-captured virions moved together with/within CD98-marked tubules in a centripetal fashion to form a VCC. We note that other viruses, including adeno-associated virus 2 (AAV2) and Simian Virus 40 (SV40), utilize CLIC/GEEC endocytosis to enter cells [52, 53]. Endocytosis of cargo through CLIC/GEEC is characterized by the formation of plasma membrane invaginations that form extended tubules, reminiscent of the tubules previously observed connecting to the VCC of HIV-infected MDMs [51,44]. The tubules demonstrated here are very likely the same tubules previously seen by ion abrasion scanning EM in infected macrophages [23]. We suggest from this work that virion-containing tubules seen in these static images in infected macrophages represent Siglec-1-captured viruses that are moving centripetally from the PM toward the VCC, rather than virions leaving the VCC to exit the cell.

The requirement for both dynamin-2 and the CLIC/GEEC pathway for internalization of HIV-1 particles and formation of the VCC adds some complexity to this pathway, as CLIC/GEEC cargo are typically dynamin-independent. A parallel may be found in Cholera toxin subunit B (CTxB), which also relies on both the CLIC/GEEC pathway and dynamin for entry [54]. Although not required for CLIC formation, dynamin has been shown to be recruited to CLICs following plasma membrane scission [44]. Recruitment of dynamin to CLICs as they mature may explain the involvement of the BAR domain protein GRAF1, which is known to be a component of tubular endosomes involved in the uptake of GPI-linked proteins and glycosphingolipids [55]. GRAF1 binds to dynamin and plays a role in processing of CLIC/GEEC endocytic membranes, suggesting that dynamin itself may facilitate endocytosis of CLIC/GEEC cargo at a step following initial internalization from the plasma membrane [56]. We propose a model in which Siglec-1-mediated capture of HIV-1 particles and nanoclustering of Siglec-enriched membranes leads to tubulation of the membrane inwards to form the CLIC. The fact that both Siglec-1 and VLPs are found in CD98+ tubules that fail to progress to form a VCC following dynamin-2 depletion suggests that dynamin-2 acts at a stage following particle internalization. It is possible that dynamin's membrane scission activity is required for the formation of the VCC (as depicted in Fig 9), although this is not yet proven and remains under investigation. We note that Menager and colleagues previously found that inhibition of dynamin-2 in DCs reduced HIV-1 trans-infection, resulting in concentrations of viral particles in macropinosome-like vesicles [57]. While results shown here implicate the CLIC/GEEC pathway rather than macropinocytosis in the viral internalization pathway, our data are consistent with the prior finding that dynamin-2 contributes to trans-infection.

A role for lipid rafts in HIV-1 particle capture and uptake by DCs has been previously reported [58]. In this study, extraction of cholesterol using methyl-β-cyclodextran or filipin III reduced virus binding and uptake. CLIC/GEEC endocytosis occurs in cholesterol-rich raft-like domains of the PM, and even mild depletion of cholesterol disrupts endocytosis via this pathway, potentially connecting these prior findings to findings reported here [59, 60].

Results here provide insights into several aspects of VCC formation that have been puzzling. We previously demonstrated that VLP addition to MDMs leads to VCC formation, creating a compartment with the same markers characteristic of the VCC in infected MDMs [30]. When an "authentic" VCC first forms within infected MDMs, VLPs added exogenously are taken into this compartment, where they mix with the infectious particles in the VCC [30].

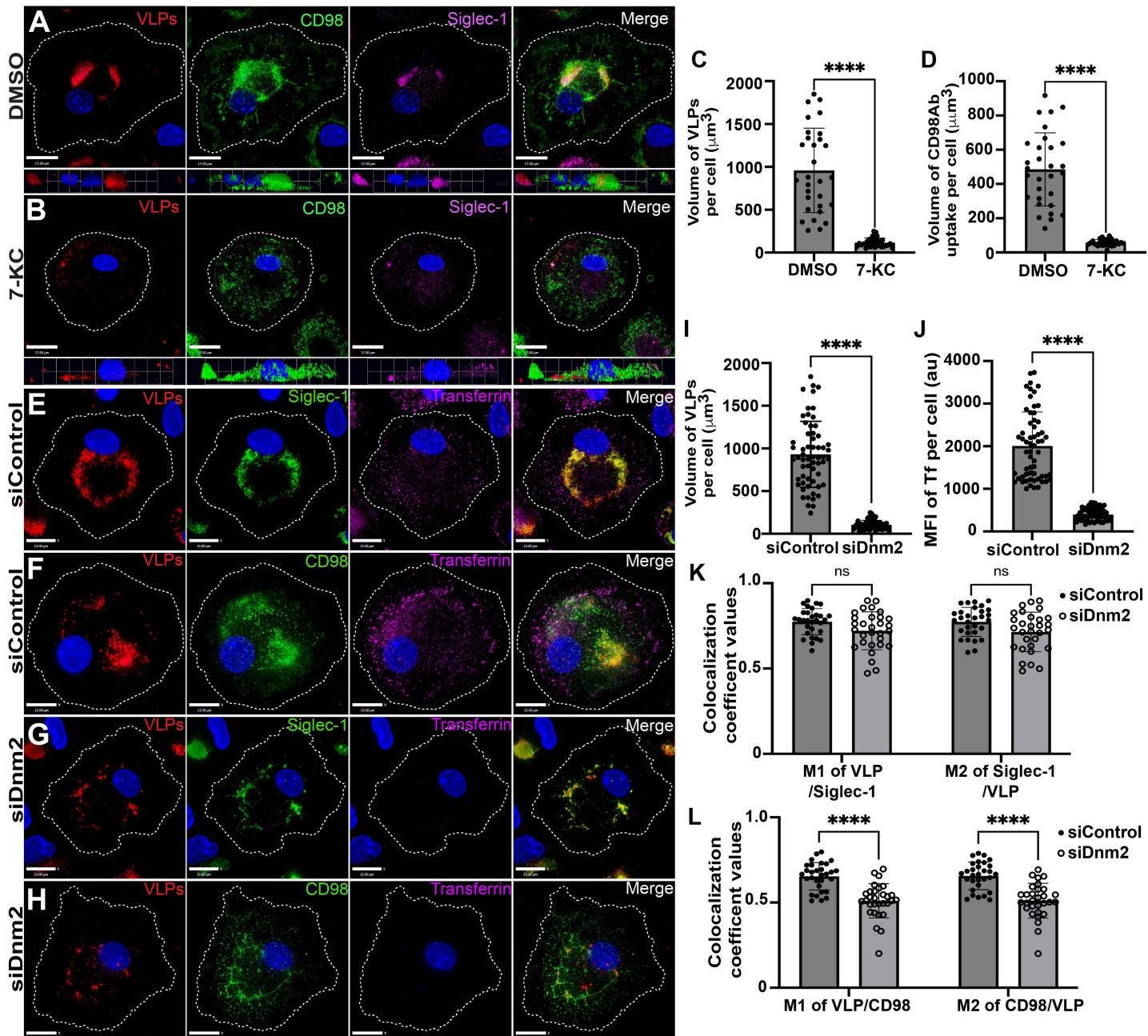

**Fig 8. Siglec-1-captured HIV-1 employs CLIC tubules and the CLIC/GEEC pathway to form the VCC in Macrophages.** (A) Control MDMs were treated with DMSO followed by addition of mCherry VLPs, with internalization for 6 hours. CD98 Ab was then added for 1.5 hours, without washing, and cells were fixed and stained. Siglec-1 staining is shown in magenta and CD98 in green. (B) MDMs were treated with 7-KC followed by VLP addition and incubation as in (A). (C) Plot of volume of VCC following DMSO or 7-KC treatment. (D) Volume of CD98+ compartment following DMSO or 7-KC treatment. (E) MDMs transfected with control siRNA, followed by VLP addition, showing the uptake of mCherry VLPs (red) together with Siglec-1 (green). Transferrin is shown in magenta. (F) Control MDMs showing the uptake of mCherry-VLPs (red) that colocalize with pulsed CD98Ab (green). (G) MDMs transfected with siDnm2, followed by VLP addition, were stained as described in (E). (H) MDMs transfected with siDnm2 were stained as described for in (F). (I) Graph of mean ± SD volume of VLPs per cell from 60 different MDMs for each experimental group. (J) Bar graph of Mean ± SD fluorescence intensity of transferrin per cell from 60 different MDMs for each experimental group. (K) Graph of mean ± SD of the colocalization coefficient values for VLPs/Siglec-1 (M1) and Siglec-1/VLP (M2) per cell from 30 different MDMs for each experimental group. (L) Graph of mean ± SD of the colocalization coefficient values for VLPs/CD98 (M1) and CD98/VLP (M2) per cell from 30 different MDMs for each experimental group. Data are representative of least three independent experiments ****p <0.001 and ns, not significant (Mann-Whitney test). Scale bar = 17 μm (A, B) and 13 μm (E-H).

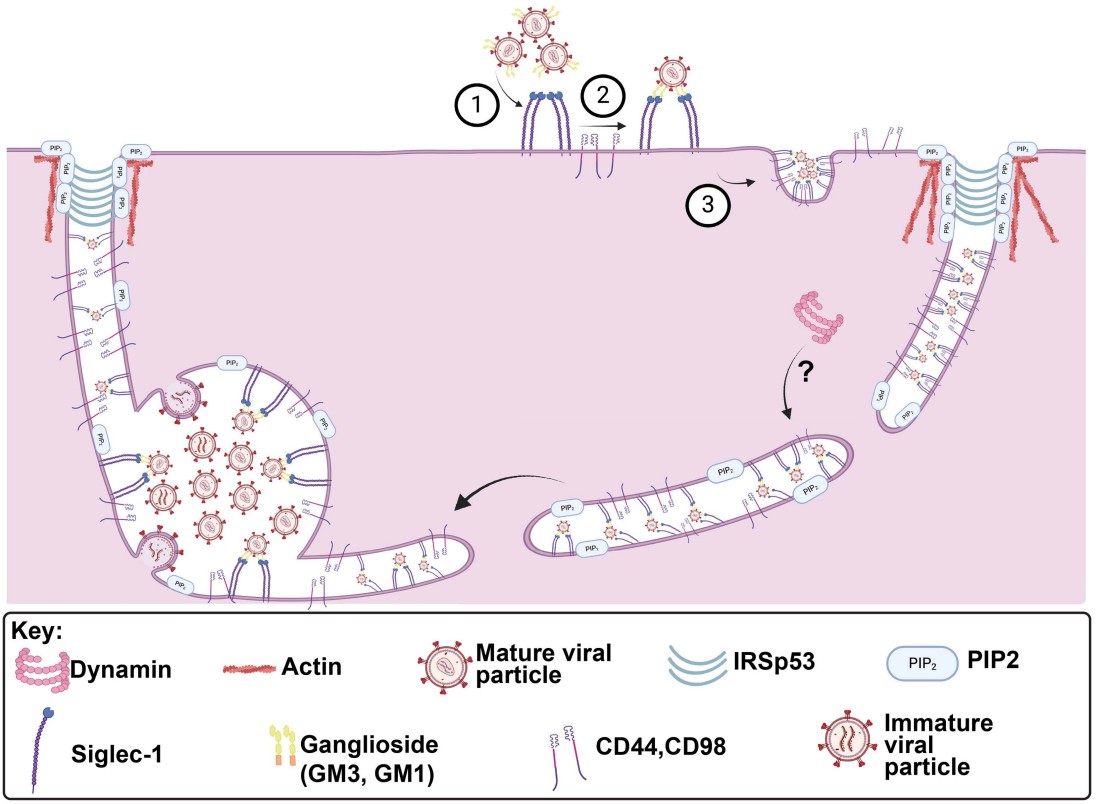

**Fig 9. Model for Siglec-1-captured HIV-1 particle uptake into MDMs to form the VCC.** Siglec-1 proteins capture HIV-1 particles on the PM by binding to gangliosides on the virion lipid envelope, followed by clustering on the cell surface. Clusters of captured HIV-1 particles are subsequently internalized via CLIC/GEEC tubules. How dynamin-2 contributes to the process is less clear (question mark), but appears to function at a later stage following tubule formation to facilitate formation of the VCC in MDMs. Created in https://BioRender.com.

This suggested that to us that particles in infected macrophages may first assemble on the PM and then be delivered into the VCC via a common pathway shared with Siglec-1-captured VLPs. However, numerous laboratories have reported assembly and budding occurring on the limiting membrane of the VCC, establishing that there is HIV-1 assembly on membrane platforms deep within the cell [17,27,50,61]. Narrow tubules extending from the VCC that sometimes reach the plasma membrane have been documented, and viruses were clearly shown within these tubules by EM [17,23,50]. By live cell microscopy, these tubules were dynamic and the connections with the surface transient [50]. Work here identifies these tubules as components of the CLIC/GEEC pathway, and we suggest that this pathway internalizes both Siglec-1-captured virions and membrane microdomain components to the VCC. Diversion of plasma membrane lipid raft components including $PI(4,5)P_2$ to the VCC through the CLIC/GEEC pathway can then set the stage for subsequent HIV assembly events to occur on VCC membranes. In support of this idea, CLIC/GEEC endocytosis occurs in regions of the PM that are enriched in $PI(4,5)P_2$ or $PI(3,4,5)P_3$, and the CLIC/GEEC regulator GRAF1 also binds to $PI(4,5)P_2$-enriched membranes [59]. VCC membranes are themselves enriched in $PI(4,5)P_2$, thus creating "assembly platforms" within the cell that have combined with late endosomal components to form the VCC [61]. The prominent presence of CLIC/GEEC cargo (CD44, CD98) in the VCC, the movement of tubular membranes together with HIV VLPs to

form a VCC, and the inhibition of VCC formation by disruption of the CLIC/GEEC endocytic pathway all support this model.

The VCC is a compartment known to facilitate trans-infection of T lymphocytes, and therefore contribute to HIV spread and pathogenesis. Disruption of CLIC/GEEC pathway or depletion of dynamin-2 in MDMs led to a significant inhibition of trans-infection, consistent with a model in which both pathways contribute to particle uptake and VCC formation, and are therefore required for efficient trans-infection. Transmission from infected MDMs to T cells is known to be an efficient mode of HIV-1 spread, occurring through transient cell-cell contacts referred to as the virological synapse (VS) [62, 63]. The formation of a VS may provide not only an efficient transmission mechanism, but may help to shield virus from neutralizing antibodies that would be effective in preventing cell free infection [64–66]. The VCC may also serve as a reservoir of infectious virions within tissue macrophages, contributing to HIV persistence in the presence of antiretroviral therapy. Identification of the important role the CLIC/GEEC pathway plays in VCC formation will facilitate future efforts to disrupt VCC formation, with the goal of inhibiting transmission of HIV-1 from macrophages to uninfected target cells and contributing to the elimination of the macrophage reservoir.

## Materials and methods

### Ethics statement

Human blood for the preparation of monocyte-derived macrophages and other experiments in this work was obtained from volunteer donors and was de-identified prior to handling by the investigators. Written informed consent was obtained from participants. Blood was collected under a protocol approved by the Cincinnati Children's Hospital Institutional Review Board.

### Isolation and maturation of MDMs

Human peripheral blood mononuclear cells (PBMCs) were isolated from human blood using Ficoll-Hypaque gradient centrifugation. PBMCs from the buffy coats were pooled and extensively washed with PBS. Monocytes were enriched by negative selection method using Miltenyi pan monocyte isolation kit (Miltenyi Biotec Inc). Enriched monocytes were plated on either multiwell chambered coverglass (Cellvis) or poly-D-lysine coated 35mm³ MatTek dishes (MatTek Corporation) or poly-D-lysine coated 6 well plates (Corning) or poly-D-lysine coated 12 well plates (Corning) and cultured in RPMI 1640 media supplemented with 10% FBS, 100 U/ml penicillin, 100 ug/ml streptomycin and 2mM Glutamine (macrophage RPMI medium). The cells were matured in the presence of 5 ng/ml GM-CSF (PeproTech Cat no 300-03) for 7 days to facilitate maturation into monocyte derived macrophages (MDMs). Following maturation, MDMs utilized for VLP capture and internalization were stimulated with 2000 U/ml of universal Type I IFN (PBL Assay Science Cat. No. 11200-2) to upregulate Siglec-1 expression. MDMs for infection experiments were not stimulated with IFN.

### Reagents, chemicals and antibodies employed

Dyngo4a (Cat no AB120689) was obtained from Abcam and used at a concentration of 20 μM. Pitstop 2.0 (Cat SML1169) was obtained from Sigma-Aldrich and Pitstop negative control (Cat AB120688) was obtained from Abcam and used at a concentration of 5 μM. 7-Keto Cholesterol (7-KC) (Cat 16339) was obtained from Cayman Chemical Company and was used at 30 μM. Fluorescently labeled Alexa Fluor 647 and Alexa Fluor 488-labeled human transferrin were purchased from Jackson ImmunoResearch (Cat. No. 009-600-050 and 009-540-050) and employed at a dilution of 1:100. Fluorescently labeled 70 kDa Dextran (Cat D1822)

was purchased from Thermo Fisher Scientific and was used at a concentration of 100 μg/ml. Mouse monoclonal antibodies to human CD9 (Cat 312102) diluted to 1:250, human Siglec-1 (Cat 346002) diluted to 1:200 and Alexa Flour 647 anti-human CD169 (sialoadhesin, Siglec-1) (Cat 346005) diluted to 1:150 were from Biolegend. FITC-KC57 was obtained from Beckman Coulter (Cat 66046650) and was used at a dilution of 1:500. EPS15 antibody (Cat NBP1-89221) and dynamin-2 antibody (Cat NBP2-47477) were purchased from Novus Biologicals. Rabbit Polyclonal anti-human FCHO2 antibody (Cat PA5-31696) from Thermo Fisher Scientific was used for immunoblotting (dilution 1:1,000). Caveolin antibodies for western blotting included rabbit polyclonal anti-human caveolin-1 (Cat ab2910) from Abcam; Rabbit anti-human caveolin-2 (Cat 85225) from Cell Signaling, and mouse anti-human caveolin-3 (Cat MAB6706) from R&D Systems. Mouse monoclonal antibody to human CD44 (Cat ab6124) from Abcam was used in immunofluorescence studies at 1:500 dilution. Anti-human CD98 antibody (Cat 315602, 1:250) and FITC anti-human CD98 Antibody (Cat 315603, 1:150) were from Biolegend. Rabbit polyclonal anti-IRSp53 antibody (Cat ab126057) from Abcam was used for immunoblotting (1:5000) and immunofluorescence (1:500). BV421 mouse anti-human CD98 antibody (Cat 744502) used at a 1:100 dilution for pulse-chase experiments and anti-human CD3-APC (Cat 555342,1:5) from BD bioscience. Mouse (Cat Ma5-11869) and rabbit (Cat A20266-100UL) actin antibodies were purchased from Thermo Fisher Scientific and Millipore Sigma for western blotting (1:5000). Secondary antibodies included Alexa Fluor 488-conjugated goat anti-mouse, Alexa Fluor 488-conjugated goat anti-Rabbit, Alexa Fluor 647-conjugated goat anti-mouse and Alexa Fluor 647-conjugated goat anti-Rabbit (Invitrogen) at a dilution of 1:1,000.

## VLP production

HIV-1 mCherry VLPs were generated by transient transfection of HEK 293T cells with a pVRC-3900 and pVRC/ GAGOPT-mCherry at a ratio of 4:1 as previously described [30]. The HIV-1 Pr55$^{Gag}$ construct, pVRC-3900, is an expression plasmid encoding a codon-optimized HIV-1 Gag polyprotein and was kindly provided by Gary Nabel (VRC, NIH). HEK293T cells were transfected using JET prime reagent (PolyPlus). VLPs were harvested 48 hours after transfection, supernatants clarified and concentrated by centrifugation through a 20% sucrose cushion. VLPs were resuspended in ice cold serum free RPMI 1640 and the aliquots were stored at -80°. For use, the VLP aliquot was thawed and filtered through a 0.45μm syringe filter before addition to MDMs.

## Viral stock generation and MDM infection

pNLUdel proviral plasmid was obtained from Klaus Strebel, NIAID, NIH[67]. VSV-G-pseudotyped HIV-1 NLUdel stocks were generated by transfection of 293T cells using Jetprime transfection reagent (Polyplus). 36 hours after transfection, viral supernatant was harvested, filtered through a 0.45 μm filter and stored at -80 °C. The viral titer was determined using a TCID$_{50}$ assay performed in TZM-bl indicator cells (obtained through the NIH HIV Reagent Program, Division of AIDS, NIAID, NIH: TZM-bl Cells, ARP-8129, contributed by Dr. John C. Kappes, Dr. Xiaoyun Wu and Tranzyme Inc). Primary HIV-1 isolate BaL stocks were prepared as follows: Human PBMCs were resuspended in complete RPMI medium. Primary HIV-1 isolates were propagated in PBMCs stimulated with 5 μg/ml phytohemagglutinin (PHA) and 5% IL-2. The IL-2/ PHA-stimulated cells were infected using a high-titer seed stock of virus minimally passaged in PBMCs, derived from a viral stock obtained through the NIH HIV Reagent Program (HIV-1$_{BaL}$, ARP-510, contributed by Dr. Suzanne Gartner, Dr. Mikulas Popovic and Dr. Robert Gallo). 2 mL of virus was transferred to the flask containing

freshly stimulated PBMCs and incubated overnight at 37 °C in 5% $CO_2$. Cells were washed extensively and resuspended in 30 ml of RPMI-GM with IL-2. The virus-containing supernatants were collected, clarified by centrifugation, and filtered through a 0.45 μm filter and stored at -80 °C.

For titering of viral stocks, TZM-bl cells were incubated for 48 hours after infection with viral supernatant and 100 μl of supernatant was removed from each well prior to the addition of 100 μl of Britelite (Perkin-Elmer) substrate. Measurement of infectivity was performed using 150 μl of cell/substrate mixture in a black 96 well solid plate and measurement of luminescence was performed using a plate luminometer. Human MDMs were infected with the viral stock at a MOI of 3 for $NL_{Udel}$ experiments and a MOI of 0.5 for HIV-$1_{BaL}$ experiments.

### p24 ELISA.

A p24 antigen capture ELISA was used for measuring the amount of p24 found in MDMs supernatant as previously described [30]. Briefly, a hybridoma line expressing murine anti-p24 capture antibody 183-H12-5C (CA183) was acquired from Bruce Chesebro and Kathy Wehrly through the NIH AIDS Research and Reference Reagent Program and a stock of CA183 produced. Plates were coated with CA183, and lysed supernatant samples added together with dilutions of recombinant p24 to generate a standard curve. The detection of bound p24 was determined using HIV-Ig, obtained from NABI through the NIH AIDS Research and Reference Reagent Program. Colorimetric analysis was performed using the Immunopure TMB Substrate Kit (Pierce, Rockford, IL) and absorbance was read at 450 nm on a plate reader.

### Immunofluorescence staining, imaging, and image analysis

MDMs were fixed with 4% PFA in PBS for 10 min at room temperature and permeabilized with 0.1% Triton X-100 for 5 minutes, then washed thoroughly with PBS. Fixed cells were blocked with Dako blocking buffer supplemented with 1 μg/ml human IgG. Cells were then incubated with antibodies against Siglec1, CD9, CD44, CD98, IRSp53 or KC57 in Dako antibody diluent (Dako) overnight at 4 °C. Cells were washed thoroughly before incubations with appropriate secondary antibodies if not incubated with a primary labeled antibody. For visualization of the nucleus, the MDMs were stained with DAPI (4'-6'-diamidino-2-phenylindole) at 300 nM in PBS. Imaging was performed on a Deltavision RT deconvolution microscope (Applied Precision/ Leica Instruments). Data analysis for colocalization coefficients M1 and M2, volume measurements, and mean immunofluorescence (MFI) were performed using Volocity software (Perkin-Elmer/Quorum Technologies). Volocity's colocalization application was used to quantify the colocalization coefficient values of M1 and M2 of individual cells, from which statistical comparisons of multiple cells were compiled for statistical analysis. To quantify volume and MFI, MDM volumes were 3D-reconstructed, outlined using the trace region tool, and measurement functions from Volocity used determine volume and MFI for individual cells. The number of cells quantified for each experimental arm is indicated in the figure legends.

### VLP uptake in chemical inhibition experiments

VLP uptake assays were performed in GM-CSF-matured MDMs after stimulation with IFN-U. HIV-1 mCherry VLPs or both HIV-1 mCherry VLPs or HIV-1 Gag-GFP VLPs were added to MDMs in the presence of the indicated inhibitors or controls. For the experiments with Dyngo4a, the MDMs were first serum starved for 30 minutes at 37 °C prior to addition of HIV-1 Gag-GFP VLPs. After serum starvation, 600 ng of HIV-1 Gag-GFP VLPs were added to the MDMs for the next 5 hours. MDMs were washed with serum free RPMI to remove

residual HIV-1 Gag-GFP VLPs. Next, the MDMs were treated with either 20 μM of Dyngo4a or 20 μM DMSO control for 30 minutes at 37 ºC. After 30 minutes, 400 ng of HIV-1 mCherry VLPs in serum free RPMI with either 20 μM of Dyngo4a or 20 μM DMSO control were added to the MDMs for 2.5 hours. During the last 30 minutes of the HIV-1 Gag-mCherry VLP uptake, the MDMs were incubated with Alexa Fluor 647-conjugated human transferrin. After the internalization, the MDMs were rinsed with PBS and then fixed with warm 4% PFA in PBS for 10 minutes at 37 °C and were rinsed with 100 nM glycine in PBS and processed for immunofluorescence staining. For experiments containing 7-KC, MDMs were treated with either 30 μM of 7-KC or 30 μM DMSO control for 12 hours. After this treatment, 400 ng of HIV-1 mCherry VLPs in serum-free RPMI with either 30 μM of 7-KC or 30 μM DMSO control were added to the MDMs for 14 hours. During the last 30 minutes of the VLP uptake, MDMs were incubated with FCR blocker and CD44Ab or with FCR blocker and FITC-CD98Ab at 37 °C. MDMs were then washed with PBS and fixed and stained for Siglec-1.

### Off-target effect assays in MDMs

MDMs were plated on either 35 mm³ MatTek dishes (MatTek corporation) or multiwell chambered coverglass (Cellvis). After maturation, MDMs were treated with 2000 U/ml IFN-U for 2 days. MDMs were pre-incubated with inhibitors 7-KC for 12 hours, Pitstop 2.0 for 30 minutes, Dyngo4a for 30 minutes, or controls such as DMSO or Pitstop Negative control for 30 minutes in serum-free RPMI. After the pre-incubation, MDMs were treated with different markers for 30 min for internalization. CD98 Ab and CD44 Ab uptake was used as a marker for CLIC/GEEC pathway, human labeled transferrin (5 ug/ml) for clathrin-mediated endocytosis and fluorescently labeled 70kDa dextran (100 μg/ml) for macropinocytosis. MDMs were then washed with either PBS or an acidic buffer 0.2M glycine with 0.15M NaCl before fixation with 4% PFA.

### Phagocytosis assay

MDMs were treated with 2000 U/ml IFN-U for 2 days post maturation. On day 2, MDMs were treated with either 30 mM of 7-KC or DMSO for 12 hours. After the pre-incubation, the MDMs would have a change in medium to serum free RPMI that contained either 30 mM of 7-KC or DMSO with HIV-1 mCherry VLPs for 14 hours. After 14 hours, latex beads coated with fluorescently labeled IgG from the phagocytosis assay kit (Cayman Chemicals Cat. No. 500290) were added to the MDMs for 1 hour at 37 °C. Then, the MDMs were washed with assay buffer to remove any beads from the MDM cell surface. After washing, the MDMs were fixed and stained as outlined above.

### Knockdown experiments in MDMs with VLP uptake

SiCtrl (12935300), siDnm2 (4390824 ID: s4212) and siEps15 (4392420 ID: s4775) were obtained from Thermo Fisher Scientific and both siFCHO2 (Cat. sc-91916) and siIRSp53 (Cat. sc-60863) were obtained from Santa Cruz Biotechnology. For knockdown experiments, MDMs were treated with either 50 nM of siCtrl, siDnm2, siFCHO2, siIRSP53 or siEps15 using Lipofectamine RNAiMax reagent (Life Technologies) using the manufacturer's protocols. After 7 hours, an equal amount of serum containing medium was add to the transfection medium and on day 1 post-transfection, the medium removed and replaced with macrophage medium. 3 days post-transfection, MDMs treated with either siDnm2 or siEps15 received a second round of siRNA treatment, and on day 5, the si-MDMs were then treated with 2000 U/ml universal Type1 Interferon for 16 hours. On day 6 following the initial siRNA transfection, HIV-1 mCherry VLPs in serum free medium were added, and cells incubated for an

additional 6 hours in dynamin-2 Experiment and 14 hours for all the other experiments. The MDMs were then fixed and stained for immunofluorescence imaging.

## siRNA-mediated knockdown experiments in MDMs following HIV-1 infection

MDMs were infected with VSV-G pseudotyped HIV-1 NLUdel at MOI of 2 or with HIV-1$_{BaL}$ at an MOI of 0.5 for 16 hours. After HIV-1 infection, MDMs received either 50 nM of SiCtrl, siDnm2, siEps15, siFCHO2 and siIRSp53 as described above. After 7 hours, an equal amount of serum containing medium was add to the transfection medium. On day 1 post-transfection, the medium removed and replaced with macrophage medium. On day 6 post-transfection, MDMs treated with either siDnm2 or siEps15 received a second round of siRNA treatments as described earlier. MDMs infected with NL$_{Udel}$ were fixed on day 10 post-infection, and MDMs infected with BaL were fixed on day 12 post-infection.

## CD98 antibody pulse-chase in MDMs

MDMs were plated on either 35 mm³ MatTek dishes (MatTek corporation) or multiwell chambered coverglass (Cellvis). On day 6 after maturation, MDMs were treated with 2000 U/ml IFN-U for 2 days. On day 2, the MDMs were serum starved for 30 minutes at 37 °C. MDMs were then next incubated with BV421 anti-CD98 antibody against the endogenous CD98 protein for 1.5 hours at 37 °C to allow for internalization of the antibody bound CD98. After the pulse-chase with CD98 antibody, MDMs were rinsed twice with serum free RPMI medium, followed by addition of 400 ng of HIV-1 mCherry VLPs, and incubated for 30 minutes, 2 hours and 6 hours after the addition of VLPs prior to fixation. To obtain time point 0 after VLP addition, some MDMs were fixed immediately after VLP addition with warm 4% PFA in PBS for 10 minutes at 37 °C. After fixation, MDMs were rinsed with 100 nM glycine in PBS. MDMs were then permeabilized with 0.05% Triton 100 for 5 minutes and washed with 0.025% Triton 100 for 5 minutes. MDMs were blocked with Dako blocking buffer supplemented with 1 μg/ml human IgG. MDMs were then incubated with Alexa Flour 647 anti-human CD169 antibody in Dako antibody diluent (Dako) overnight at 4 °C. After overnight staining, the MDMs were washed with 0.025% Triton multiple times. For visualization of the nucleus, MDMs were stained with DRAQ5 at 1:250 dilution in PBS for 15 minutes at room temperature. For the CD98 antibody pulse-chase experiment with 7-KC inhibitor addition, MDMs were treated with either 30 μM of 7-KC or 30 μM DMSO control for 12 hours. After this, 400 ng of HIV-1 mCherry VLPs with either 30 μM of 7-KC or 30 μM DMSO control were added to the MDMs for the next 6 hours. During the last 1.5 hours of the VLPs uptake, the MDMs were incubated with BV421 anti-CD98 antibody against the endogenous CD98 protein to allow for internalization of CD98. Following the CD98 antibody pulse-chase, MDMs were fixed with warm 4% PFA in PBS for 10 minutes at 37 °C and stained for Siglec-1. For dynamin-2 depletion experiments, siRNA depletion in MDMs as outlined above. On day 6 following the first siRNA treatment, CD98 antibody pulse-chase was performed as described, followed by addition of 400 ng of HIV-1 mCherry VLPs for 6 hours. During the last 30 minutes of the 6-hour time course, transferrin was add to the MDMs. After 6 hours, MDMs were fixed with warm 4% PFA in PBS for 10 minutes at 37 °C and stained and imaged as before.

## Macrophage-T cell trans-infection

MDMs were plated on poly-D-lysine coated 12 well plates (Corning) and GM-CSF matured for 7 days. Matured MDMs were then overnight infected with HIV-1$_{BaL}$ at MOI of 1.5. The following day post-infection, MDMs were treated with siControl, siDnm2, or siIRSp53 as

described above in knockdown experiments in MDMs with HIV-1 infections. MDMs received a second round of siRNA treatments on day 4 post-infection as described earlier. siRNA-treated MDMs were left untreated or were treated with 10 μM indinavir sulfate (obtained through the NIH AIDS Reagent Program, Division of AIDS, NIAID, NIH) for 2 hours prior to co-culturing with PHA-activated autologous T lymphocytes on day 7 post-infection. For preparation of autologous T cell cultures, resting autologous T lymphocytes were activated by incubation of 5 μg/ml phytohemagglutinin A (PHA) (Millipore Sigma, Cat. No. L1668-5MG) and 100 U/mL of recombinant Human IL-2 Protein (R&D Systems, Cat. No. 202-IL-050) for 3 days in RPMI 1640 media supplemented with 20% FBS, 100 U/ml penicillin, 100 ug/ml streptomycin and 2mM Glutamine (complete RPMI medium). After 3 days, cells were washed to remove PHA and were maintained in media supplemented with IL-2. T lymphocytes were added to MDM cultures at ratio of 1:3 (MDM/T cells) in macrophage medium, and cells were co-cultured for 12 hours at 37 °C. T lymphocytes were then separated from MDMs using Versene solution (Thermo Fisher Scientific), washed in serum-free RPMI-1640 and cultured for 24 hours in complete medium with 5 ng/ml rhIL-2 (R&D Systems) and 10 μM indinavir sulfate, prior to labeling with Zombie Violet Fixable Viability (Biolegend Cat. No. 423113) for 15 minutes at room temperature. Cells were then washed and fixed with 4% PFA for 10 minutes, and stained with anti-human CD3-APC (BD Biosciences, Cat No. 555342) for 1 hour at 4 °C. Following cell surface staining, cells were permeabilized using Fixation/Permeabilization Solution Kit (BD Biosciences). The cells were then stained for intracellular p24 using anti-HIV-1 p24 KC57-FITC for 1 hour at 4 °C. Cells were analyzed using a Cytek Northern Light flow cytometer, and data analyzed using FlowJo (Treestar Inc.). The gating strategy used to determent the percentage of viable CD3+ and p24+ cell is shown in S7C Fig.

## Western blotting

MDM lysates were prepared by lysing in RIPA buffer containing protease inhibitors for 30 min on ice, followed by clarification by centrifuging at 15000 RPM on a tabletop centrifuge at 4 °C for 30 min. Protein concentration of the lysates was measured using the DC protein assay kit (BioRad 5000111) following the manufacturer's instructions. 40 μg of total protein was loaded onto precast 4-12% Bis-Tris NuPAGE gels (Thermo Fisher Scientific). Protein was transferred to nitrocellulose membrane using the semi-dry trans Blot transfer (BioRad) following manufacturer's instructions. The membrane was blocked, probed with appropriate primary and secondary antibodies, and imaged using the LiCOR infrared imaging system.

## Library Preparation and Sequencing for Bulk RNA seq

Raw reads from purified RNA of cultured macrophages were processed using the nf-core/rnaseq pipeline version 3.4 [https://dx.doi.org/10.1038/s41587-020-0439-x]. Briefly, adapter sequences and low-quality reads were filtered and trimmed using FastQ and Trim Galore. Filtered reads were mapped to the human GRCh38 reference genome version 108 using STAR [https://dx.doi.org/10.1093/bioinformatics/bts635]. Duplicate reads were removed using MarkDuplicates. Next, Salmon [https://dx.doi.org/10.1038/nmeth.4197] was used to generate a count matrix from genes with mapped reads. Normalized counts were then created by normalizing raw counts to the library size. The full description of the pipeline is available at https://nf-co.re/rnaseq.

## Statistical analysis

All of the graphical data plotted in the figures are presented as mean ± Standard Deviation (SD) using GraphPad Prism version 10.0. Statistical significance between two groups were

determined by using Mann-Whitney t-test. The statistical significance between 3 groups in Fig 2 graphs used an Kruskal-Wallis ANOVA test, followed by Dunnett's multiple comparisons test. The statistical significance between 3 groups in Figs 6 and S7 graphs used the Brown-Forsythe and Welch ANOVA test followed by Dunnett's multiple comparisons test.

## Supporting information

**S1 Fig. Examination of off-target effects of Dyngo4a and Pitstop 2.0 on the CLIC/GEEC pathway and macrocytosis in MDMs and 3D view of dynamin-2 knockdown in BaL infected MDMs** . (A-D) MDMs were treated with either 20 μM DMSO or Dyngo4a for 30 minutes. Transferrin and CD98 Ab or Transferrin and 70kD Dextran internalized for 30 minutes before fixation of MDMs. MDMs were washed, fixed in 4% PFA and, and stained with DAPI. (E-F) MDMs were treated with either 20 μM Pitstop negative control or Pitstop 2.0 for 30 minutes. MDMs then internalized transferrin and CD98 Ab for 30 minutes before fixation. Cells were washed, fixed in 4% PFA and, stained with DAPI. Scale bar = 5.40 μm. Data are representative of least three independent experiments. (G-J) 3D side view of the images from Fig 1G-1J.
(TIF)

**S2 Fig. p24 production and release from dynamin-2, IRSp53 or control siRNA treated infected MDMs** . p24 production and release from dynamin-2, IRSp53 or control siRNA treated and HIV-1$_{BaL}$-infected MDMs was assessed over 12 days using a p24 ELISA. The efficiency of particle production is plotted as mean ± SD of the extracellular p24 over the 12 days of infections from each of the different donor used this experiment. Arrows indicate 3 timepoints when the medium was changed, resulting in a transient drop in the amount of extracellular p24.
(TIF)

**S3 Fig. 3D view of the CME knockdown experiments.** (A-C) 3D view of the images from primary Fig 2 evaluating VLP uptake into VCC following siRNA treatments. S3A corresponds to panel A in Fig 2, S3B corresponds to panel D in Fig 2, and S3C corresponds to panel E in Fig 2. (D-F) 3D view of the images from primary Fig 2 evaluating VCC formation in infected MDMs following siRNA treatments. S3D corresponds to panel H from Fig 2, S3E corresponds to panel I from Fig 2, and S3F corresponds to panel J from Fig 2.
(TIF)

**S4 Fig. Caveolin Expression in human MDMs and HeLa cells.** (A) Western blot for caveolin1. (B) Western blot of caveolin 2. (C) Western blot of caveolin 3.
(TIF)

**S5 Fig. 3D view and colocalization coefficient values of CLIC/GEEC Cargo in the VCC.** (A-D) 3D view of the images from primary Fig 3A-D. S3A corresponds to Fig 3A, S3B corresponds to Fig 3B, S3C corresponds to Fig 3C, and S3D corresponds to Fig 3D. (E) Colocalization coefficient values for p24/CD98 (M1) and CD98/p24 (M2).
(TIF)

**S6 Fig. Examination of potential off-target effects of 7-KC treatment.** MDMs were treated with 30 μM DMSO (A and B) or 7-KC (C and D) for 12 hours. Transferrin and CD98 Ab or CD44 Ab were added and internalized for 30 minutes before fixation. MDMs were washed, fixed in 4% PFA and stained with DAPI. Scale bars = 8.00 μm. (E) Volume of transferrin uptake per cell (n.s = P value of >0.05 by Mann-Whitney Test). (F) Volume of CD44 Ab uptake per cell (μm³), mean with SD. (G) Volume of CD98 Ab uptake per cell (μm³), mean ± SD. (H and I) MDMs were treated with 30 μM DMSO or 7-KC for 12 hours, followed by

addition of mcherry HIV-1 VLPs and incubation for 14 hours. FITC-labeled IgG-opsonized latex beads were internalized for 1 hour. Scale bars= 9.00 μm. (J) MFI ± SD of opsonized beads per cell (n.s = P value of >0.05 by Mann-Whitney test). (K) Volume of VLP uptake per cell with the mean ± SD. (****P value of <0.0001) (Mann-Whitney test). (L) Percentage of live cells after 12 hours of treatment as determined by Zombie live/dead indicator staining (BioLegend) and analyzed via flow cytometry (mean ± SD) (Mann-Whitney test). Data are representative of least three independent experiments.
(TIF)

**S7 Fig: Knockdown and gating strategy for trans-infection experiments.** (A) Representative knockdown of Dnm2 in MDMs that were transfected with either siControl or siDnm2, as shown by western blot. (B) Representative knockdown of IRSp53 in MDMs that were transfected with either siControl or siIRSp53, as shown by western blot. (C) The gating strategy used to determent the percent of live CD3+ and p24+ T-cell. (D) Flow cytometry plots gated for CD3+ and p24+ from Donor 2 showing the percent of HIV-1 transmission in T-cells after indinavir starting day 3 post-infection. (E) Bar graph of the mean ± SD of percent of CD3+ p24+ cells, where all MDMs were treated with indinavir from day 3 post-infection. Data from 3 donors, with 3 technical replicates each.
(TIF)

**S8 Fig. 3D representation of 6 hour image from Fig. 7 and colocalization over time.** (A) 3D representation of the indicated VCC components from Fig. 7 at 6 hour time point. (B) M1 and M2 colocalization coefficients for VLPs and CD98 antibody over time.
(TIF)

**S9 Fig. 3D view of control and dynamin-2 knockdown from Fig. 8.** (A) 3D representation of overlay image from Fig 8E. (B) 3D representation from Fig. 8F. (C) 3D representation from Fig. 8G. (D) 3D representation from Fig. 8H.
(TIF)

**S1 Table. Caveolin RNA expression by RNAseq**. Normalized counts as transcripts per million (TPM) are shown for CAV1, CAV2, and CAV3 from MDMs, with controls of housekeeping genes GAPDH and ACTB (beta actin). MDMs were infected with HIV-1 BaL at MOI of 0.25, and samples reflect day 0 (uninfected) and post-infection days 1, 2, 4, 6, and 8. Measurements are in triplicate.
(XLSX)

**S1 Data. Quantitative data used in calculations corresponding to primary figures**. Numeric values used to generate graphs, means, and standard deviations for primary figures are included on tabs, with each tab indicating the relevant figure panel.
(XLSX)

**S2 Data. Quantitative data used in calculations corresponding to supplemental figures**. Numeric values used to generate graphs, means, and standard deviations for supplemental figures are included on tabs, with each tab indicating the relevant figure panel.
(XLSX)

## Acknowledgments

We thank Matthew Kofron of the CCHMC Bio-imaging and Analysis Facility (RRID: SCR_022628) for helpful input regarding image analysis. We would like to acknowledge the assistance of the Research Flow Cytometry Facility (RRID: SCR_022635) at Cincinnati Children's Hospital Medical Center.

## Author contributions

**Conceptualization:** Paul Spearman.

**Data curation:** Lingmei Ding.

**Formal analysis:** Kathleen Candor, Paul Spearman.

**Funding acquisition:** Paul Spearman.

**Investigation:** Kathleen Candor, Lingmei Ding, Sai Balchand.

**Methodology:** Kathleen Candor, Lingmei Ding, Sai Balchand, Jason Hammonds, Paul Spearman.

**Project administration:** Paul Spearman.

**Resources:** Jason Hammonds, Paul Spearman.

**Supervision:** Paul Spearman.

**Validation:** Kathleen Candor, Paul Spearman.

**Writing – original draft:** Kathleen Candor, Paul Spearman.

**Writing – review & editing:** Kathleen Candor, Jason Hammonds, Paul Spearman.

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
