## [Decision Letter · Decision Letter 0]

9 Oct 2024

Dear Prof. Spearman,

Thank you very much for submitting your manuscript "The CLIC/GEEC pathway regulates particle endocytosis and formation of the virus-containing compartment (VCC) in HIV-1-infected macrophages" for consideration at PLOS Pathogens. As with all papers reviewed by the journal, your manuscript was reviewed by members of the editorial board and by three independent reviewers with complementary expertise. In light of the reviews (below this email), we would like to invite the resubmission of a significantly-revised version that takes into account the reviewers' comments.

We cannot make any decision about publication until we have seen the revised manuscript and your response to the reviewers' comments. Your revised manuscript is also likely to be sent to reviewers for further evaluation.

Sincerely,

Li Wu, PhD

Academic Editor

PLOS Pathogens

Susan Ross

Section Editor

PLOS Pathogens

Michael Malim

Editor-in-Chief

PLOS Pathogens

orcid.org/0000-0002-7699-2064

Reviewer's Responses to Questions

**Part I - Summary**

Reviewer #1: In this paper, Candor et al. identify novel molecular players involved in the trafficking route of HIV-1 captured via Siglec-1 in macrophages, which are directed to the same VCC where newly synthetized viral particles bud after HIV-1 infection. These findings add to prior observations published by this team that now expand to gain insights into the molecular determinants regulating the events that lead to VCC formation. The study is carefully executed as it involves the use of drugs that are tested in functional assays to detect any putative off-target effects, which are always validated using alternative approaches, such as silencing RNAs for key involved proteins. This is therefore a well conducted study that adds novel findings to the current knowledge of how VCCs are formed in macrophages. The study lacks however trans-infection assays and re-evaluation of some conclusions bearing in mind prior studies.

Reviewer #2: The authors previously demonstrated the role of Siglec-1 in the formation of virus-containing compartments (VCCs) and HIV-1 transfer from macrophages to T cells in 2017 (PMID: 28129379). Siglec-1 capture and endocytose external HIV-1 particles, leading to the development of VCCs within HIV-infected cells. Here, they describe that the endocytosis of particles into the VCCs occurs independently of clathrin but requires dynamin and the CLIC/GEEC pathway, which colocalizes with Siglec-1 and HIV-1 particles inside the VCC. Thus, the authors suggest that after Siglec-1 captures the virus, the particles are internalized via an endocytic route involving both the CLIC/GEEC pathway and dynamin-2. They show this in the context of macrophages, using monocyte-derived macrophages (MDMs) and wt viruses for HIV-1 infection. Their results support a model in which the internalization of HIV-1 particles, along with CLIC/GEEC membranes, contributes to the formation of VCCs and tubules in HIV-infected macrophages. The experiments are mostly well conducted, the paper is well written, follows a coherent logic and the results are convincing. However, this work lacks novel approaches compared with what the authors have previously published before. Overall, this manuscript needs some improvements for publication, especially deeper mechanism(s), different representations of the data (e.g. volume of VCC has to be represented in 3D) and study of the role of CLIC/GEEC pathway in HIV-1 transfer to T cells (from macrophages and potentially from dendritic cells).

Reviewer #3: The Spearman lab previously reported that the so-called virus-containing compartment (VCC) in HIV-infected macrophages is formed through the capture of ganglioside-containing virus-like particles (VLP) by the cell surface lectin Siglec-1, which leads to their internalization. In the present study, the Spearman lab shows that although the formation of the VCC in MDMs depends on the endocytic scission factor dynamin, clathrin-mediated endocytosis is not required. Instead, the authors find that the internalization of HIV VLP is largely abolished by 7-keto-cholesterol and by knocking down IRSp53, both of which inhibit the clathrin-independent CLIC/GEEC endocytic pathway.

Altogether, the conclusions are significant and supported by the data, the data are clearly presented, and the paper is well-written.

**Part II – Major Issues: Key Experiments Required for Acceptance**

Reviewer #1: 1. The authors should carefully consider using a different term than “endocytosis” in the context of this study (including the title), as the VCC they refer to remains attached to the extracellular space and is therefore not formally “endocytosed”. This has been a long debate in the field, that was finally clarified by the pioneering study of the laboratory of Kräusslich in 2007 (PMID: 17381240). A similar polemic was later settled in the DC studies as well. For the sake of clarity, the field could greatly benefit from distinguishing this entry process from classical endocytic routes that lead to vesicle or endosome formation, where a real excision of the membranes implies disconnection from the extracellular space. My suggestion will be to use the concept of viral entry into a plasma membrane derived compartment, and avoid the term endocytosis, as viruses internalized via engulfed membranes in macrophages never end up budding off to form a discrete vesicle.

2. While the involvement of Dynamin 2 in VCC formation is clearly shown by the silencing experiments performed in Figure 1, it is less clear to me what is the actual role of this molecule in the formation process. The proposed activity on tubule excision shown in Figure 7C is not supported by the observations presented in this paper and somehow leads to the striking suggestion that scission is required to form a compartment that remains connected to the extracellular space. This aspect needs further experimental validation or clarification.

Interestingly, prior studies involving DCs identified DNM2 as a key molecule whose knockdown increased HIV-1 uptake but decreased HIV-1 trans-infection, what could be rescued by inhibiting micropinocytosis https://doi.org/10.1016/j.cell.2015.12.036. DNM2 was found to stabilize cortical actin and limit HIV-1 entry. DNM2 was also recognized as “a regulator of the actin network found co-localizing with actin-rich structures such as podosomes, actin comet tails, phagocytic cups, dynamic cortical ruffles, and lamellipodia”. Thus, it would be critical to show the consequences of DNM2 knockdown on micropinocytosis and explore if DNM2 knockdown could interfere on 1) HIV-1 VLP uptake in macrophages, 2) have an impact in micropinocytosis and 3) rescue HIV-1 trans-infection. This could help to depict a model for DNM2 in Figure 7C based on generated evidence.

3. Given the involvement of DNM2 in VCC formation, is there a role for caveolin in this process? Can “lipid raft” dependent pathways also influence Siglec-1 mediated internalization of HIV-1 as previously shown for DCs? (https://doi.org/10.1128/jvi.77.23.12865-12874.2003)

4. Figures 3, 6 and 7 lack associated quantification and do not disclose how many experiments are analyzed. These studies should be done at least in 3 independent experiments and quantified as in the rest of the figures of the paper.

5. The paper could increase relevance by addressing the impact of inhibiting the CLIC/GEEC pathway in HIV-1 trans-infection from macrophages to bystander target cells. This is critical to support the concept of interventional strategies aimed at limiting this potential viral reservoir, as suggested by the authors in the discussion section.

Reviewer #2: Main comments:

1/ Most of the images are of good qualitiy but should be complemented by z stack (as supplemental movies for example) and 3D reconstructions are necessary, in particular because most of the quantifications rely of VCC volume. The images in 2D are not sufficient for the reader to assess the effects of drugs and siRNA on the 3D volume of VCCs. It needs to be improved in the whole manuscript.

2/ It would be more informative (compared to the previous paper from the same team) to go further into the mechanism(s) and to study for example the role of microtubules or transport proteins. The role of GRAF1 mentioned in the discussion could also be tested.

3/ The role of CLIC/GEEC pathway in HIV-1 transfer to T cells should be tested as a functional assay.

4/ What is happening in Dendritic Cells ? The VCCs in these cells are supposed to be slightly different compared to the ones in macrophages. Whether the described mechanisms of VCC formation are similar in DCs (or not) could be studied.

Reviewer #3: 1. In addition to monitoring the uptake of VLP lacking Env (which they previously showed is not required for VCC formation), the authors infected MDM with replication-competent, macrophage-tropic HIV-1 and evaluated VCC formation on day 12 post infection. The interpretation of the results obtained with the latter approach is less straightforward, because it allowed multiple cycles of virus replication. It is possible that virus replication over several cycles was inhibited by knocking down dynamin 2 or IRSp53, which in turn could have indirectly affected VCC formation. Indeed, it has been reported that IRSp53 assists in the assembly of HIV-1 particles. For this reason, it appears important that the authors clarify whether virus replication was affected under their conditions.

2. Although the CLIC/GEEC pathway is typically dynamin-independent, as the authors note, they conclude that dynamin is required for VCC formation based on chemical inhibition or depletion of dynamin 2. However, the dynasore analog may have had off-target effects, and the Dyn2 depletion experiment was only done using the spreading infection approach. To clarify the role of dynamin, the effects of Dyn2 depletion on VCC formation should also be examined using the more readily interpretable VLP addition approach.

**Part III – Minor Issues: Editorial and Data Presentation Modifications**

Reviewer #1: Given that the high avidity binding of HIV-1 via Siglec-1 is due to the action of multiple receptors that interact with thousands of sialylated gangliosides, Figure 7C could represent this avidity interaction in the figure showing multiple receptors clustering with several viruses.

Reviewer #2: The western blots for inhibition of proteins need to be quantified for at least 3-4 donors (Figure 1F, 2A-B, 5C, .. ) and represented as a graph.

Why VSVG pseudotyped Vpu-deleted virus is used sometimes (Figure 2)? The use of wt viruses (Bal for instance) would be better or this should be justified.

Figure 1E is complicated to understand, the same for 4I, please explain the legend in another way

Please also provide associated videos to Figure 6

Reviewer #3: (No Response)

PLOS authors have the option to publish the peer review history of their article (what does this mean? ). If published, this will include your full peer review and any attached files.

**Do you want your identity to be public for this peer review?** For information about this choice, including consent withdrawal, please see our Privacy Policy .

Reviewer #1: No

Reviewer #2: **Yes: ** Christel Verollet

Reviewer #3: No
---

## [Decision Letter · Decision Letter 1]

12 Feb 2025

Dear Prof. Spearman,

We are pleased to inform you that your manuscript 'The CLIC/GEEC pathway regulates particle uptake and formation of the virus-containing compartment (VCC) in HIV-1-infected macrophages' has been provisionally accepted for publication in PLOS Pathogens.

Overall, three reviewers were satisfied with your revision. Reviewer 3 had two minor suggestions that you can consider revision. Please see below detailed comments. 

Best regards,

Li Wu, PhD

Academic Editor

PLOS Pathogens

Susan Ross

Section Editor

PLOS Pathogens

Sumita Bhaduri-McIntosh

Editor-in-Chief

PLOS Pathogens

orcid.org/0000-0003-2946-9497

Michael Malim

Editor-in-Chief

PLOS Pathogens

orcid.org/0000-0002-7699-2064

Reviewer Comments (if any, and for reference):

Reviewer's Responses to Questions

**Part I - Summary**

Reviewer #1: The authors have addressed all the comments raised by this reviewer and have provided additional and highly physiological models to prove the role of CLIC/GEEC in HIV-1 trans-infection mediated by MDM.

Reviewer #2: The authors have made major modifications to the manuscript, carrying out all the experiments requested.

Thus, this article is greatly improved thanks to the revision process.

Reviewer #3: Overall, this is an interesting paper, and I am satisfied with the response to my major concerns.

However, the “Response to Reviewers’ Comments” does not seem to address the minor issues that I had raised. Perhaps those minor issues got lost somehow, so I have listed them again below.

**Part II – Major Issues: Key Experiments Required for Acceptance**

Reviewer #1: (No Response)

Reviewer #2: I have no major issue

Reviewer #3: (No Response)

**Part III – Minor Issues: Editorial and Data Presentation Modifications**

Reviewer #1: Authors may consider to change the title of figure 4 to "7-KC treatment prevents VLP uptake to the VCC"

Reviewer #2: I have no minor issue

Reviewer #3: 1. The authors should explain why interferon was used (presumably to induce SIGLEC-1 expression).

2. Line 387: “ Remarkably, we found the SIGLEC-1 captured VLPs concentrated …”. Strictly speaking, the role of SIGLEC-1 was not examined in the present study, and is just inferred from previous reports. This should be reflected in the wording.

PLOS authors have the option to publish the peer review history of their article (what does this mean? ). If published, this will include your full peer review and any attached files.

**Do you want your identity to be public for this peer review?** For information about this choice, including consent withdrawal, please see our Privacy Policy .

Reviewer #1: **Yes: ** Nuria Izquierdo-Useros

Reviewer #2: **Yes: ** Dr. Christel Vérollet

Reviewer #3: No

---

## [Editor Report · Acceptance letter]

Dear Prof. Spearman,

We are delighted to inform you that your manuscript, "The CLIC/GEEC pathway regulates particle uptake and formation of the virus-containing compartment (VCC) in HIV-1-infected macrophages," has been formally accepted for publication in PLOS Pathogens.

Best regards,

Sumita Bhaduri-McIntosh

Editor-in-Chief

PLOS Pathogens

orcid.org/0000-0003-2946-9497

Michael Malim

Editor-in-Chief

PLOS Pathogens

orcid.org/0000-0002-7699-2064